# Photothermal $CO_2$ conversion to ethanol through photothermal heterojunction-nanosheet arrays

Xiaodong Li [1,6], Li Li [2,6], Xingyuan Chu[3], Xiaohui Liu[3], Guangbo Chen [3], Quanquan Guo [1], Zhen Zhang [4], Mingchao Wang [3], Shuming Wang[2], Alexander Tahn[5], Yongfu Sun [2] ✉ & Xinliang Feng [1,3] ✉

Photothermal $CO_2$ conversion to ethanol offers a sustainable solution for achieving net-zero carbon management. However, serious carrier recombination and high C-C coupling energy barrier cause poor performance in ethanol generation. Here, we report a $Cu/Cu_2Se-Cu_2O$ heterojunction-nanosheet array, showcasing a good ethanol yield under visible–near-infrared light without external heating. The Z-scheme $Cu_2Se-Cu_2O$ heterostructure provides spatially separated sites for $CO_2$ reduction and water oxidation with boosted carrier transport efficiency. The microreactors induced by $Cu_2Se$ nanosheets improve the local concentration of intermediates ($CH_3^*$ and $CO^*$), thereby promoting C-C coupling process. Photothermal effect of $Cu_2Se$ nanosheets elevates system's temperature to around 200 °C. Through synergizing electron and heat flows, we achieve an ethanol generation rate of 149.45 $\mu$mol $g^{-1}$ $h^{-1}$, with an electron selectivity of 48.75% and an apparent quantum yield of 0.286%. Our work can serve as inspiration for developing photothermal catalysts for $CO_2$ conversion into multi-carbon chemicals using solar energy.

Carbon peak and carbon neutrality are essential goals for tackling global climate change and environmental concerns[1]. Utilizing renewable energy to convert $CO_2$ into valuable chemicals plays a crucial role in achieving effective net-zero carbon management[2]. The generation of high-value-added multi-carbon ($C_{2+}$) products is highly desired for practical applications, yet it remains a formidable challenge[3]. Especially, ethanol holds a distinct advantage due to its elevated energy density (26.8 MJ $kg^{-1}$) and convenience for transportation and storage[4]. Despite the thermodynamically favorable nature of the $CO_2$ reduction to ethanol, substantial energy input is required to activate the $CO_2$ molecule (C = O bond energy, 806 kJ $mol^{-1}$)[5]. Conventional thermocatalysis requires high temperatures (200–400 °C) to achieve a

sufficient reaction rate for ethanol generation. This normally necessitates the inclusion of an additional heating device, leading to increased implementation expenses, fuel costs, and $CO_2$ emissions[6,7]. Furthermore, the heating affects the entire reaction setup, resulting in energy wastage in regions devoid of catalysts. In this scenario, employing photothermal materials to locally heat the catalytic area while harnessing photo-excited carriers for catalytic reactions would indeed offer dual benefits. Through photocatalytic processes, heat can effectively enhance the transport of excited electrons, enabling them to surpass the reaction energy barrier[8].

Although photothermal catalysis is recognized to amalgamate the advantages of traditional thermal catalysis and photocatalysis[9,10], there

[1]Max Planck Institute of Microstructure Physics, Weinberg 2, Halle 06120, Germany. [2]Hefei National Research Center for Physical Sciences at Microscale, University of Science and Technology of China, 230026 Hefei, P. R. China. [3]Faculty of Chemistry and Food Chemistry & Center for Advancing Electronics Dresden (CFAED), Dresden University of Technology, Dresden 01062, Germany. [4]School of Chemistry and Materials Science, University of Science and Technology of China, 230026 Hefei, P. R. China. [5]Dresden Center for Nanoanalysis (DCN), Dresden University of Technology, Dresden 01069, Germany. [6]These authors contributed equally: Xiaodong Li, Li Li. ✉e-mail: yfsun@ustc.edu.cn; Xinliang.Feng@tu-dresden.de

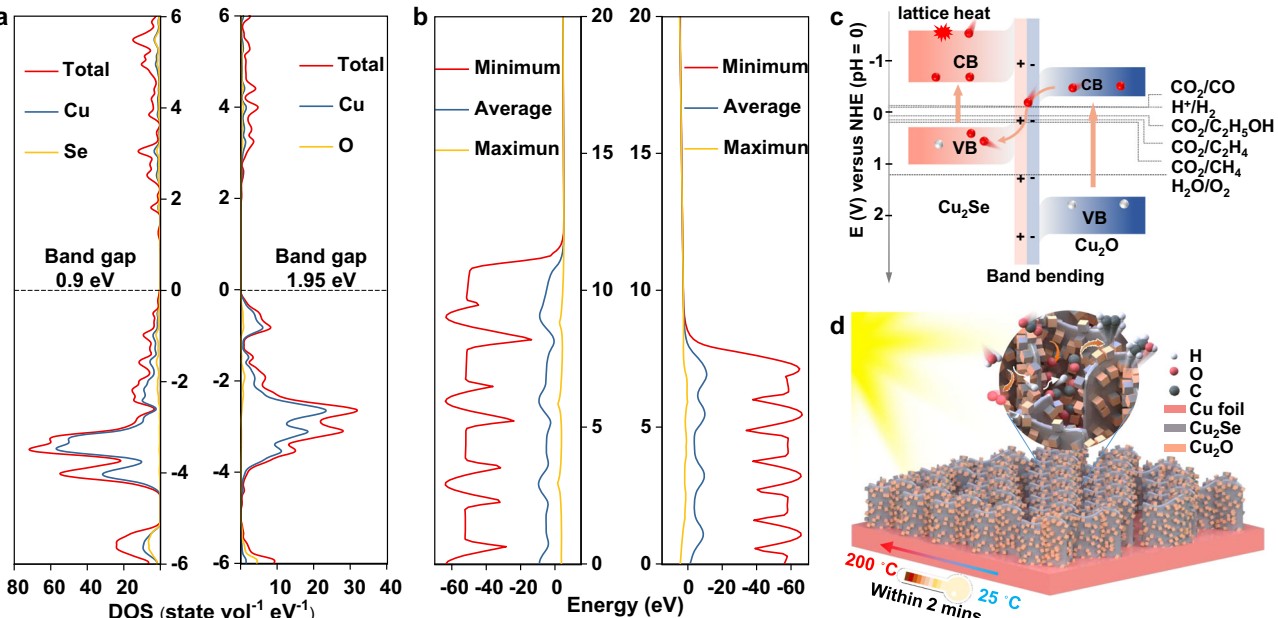

**Fig. 1 | Theoretical energy band structure of Cu-CSCO HNA. a** DOS of $Cu_2Se$ slab (left) and $Cu_2O$ slab (right); (**b**) electrostatic potential of $Cu_2Se$ slab (left) and $Cu_2O$ slab (right); (**c**) band structure alignment of $Cu_2Se$ and $Cu_2O$ heterostructure, in which Z-scheme heterostructure is formed after contacting of these two materials. Lattice heat is produced by interaction between lattice and electrons during their transition from high-energy conduction band to CBM. Band bending is induced by the built-in electric field due to the interface electron transfer. Red balls: electrons; white balls: holes. **d** Schematic diagram for photothermal $CO_2$ reduction to multi-carbon products over this array system, in which the local high temperature and well-defined intermediates (CO*, $CH_3$*) as well as the efficient-separated carriers synergistically foster the C−C coupling processes, conducive to the generation of multi-carbon products[26]. Source data are provided as a Source Data file.

are still serious limitations when applying a single photothermal material for $CO_2$ conversion. First, the heat generated by these photothermal materials can readily dissipate through infrared heat radiation, hampering the maintenance of localized high temperatures[11]. Second, to enhance light absorption capacity, photothermal materials are required to possess an intrinsic narrow-band gap (typically <1 eV)[12]. This property limits their utility for concurrent $CO_2$ reduction and $H_2O$ oxidation, as an energy difference of approximately 1.2 eV is necessary for these reactions to proceed[13]. Consequently, the concept of heterojunction systems would provide a promising alternative[14]. Third, conventional photothermal composites often rely on the plasmon effect to convert light to heat, necessitating the use of precious metals (Au, Ru, Ir), and suffering from low efficiency in the generation of hot electrons[15]. Fourth, most studies utilize $H_2$ as the reduction gas, with limited progress on utilizing $H_2O$ as the proton source for efficient $CO_2$ photothermal conversion, especially towards the generation of $C_{2+}$ products[16]. Hence, a pressing need exists for the development of strategies and catalyst configurations that can enable the conversion of $CO_2$ into liquid products, such as ethanol, especially under pure solar spectrum conditions.

To address the above challenges, herein, we report an in situ thermally enhanced approach based on a $Cu/Cu_2Se$-$Cu_2O$ heterojunction-nanosheet array (Cu-CSCO HNA) catalyst system. With our catalyst design (Fig. 1d), 1) Cu foil serves as the substrate to maintain the catalyst temperature and ensures the uninterrupted advancement of the catalytic reaction, because Cu metal is known as an exceptional infrared heat radiation barrier material[17]. 2) Vertically in situ grown two-dimensional (2D) $Cu_2Se$ nanosheets induce the photothermal effect and act as the catalyst for $CO_2$ conversion due to their abundant Cu active sites and the theoretical 1 sun-heating temperature of 271 °C[17]. 3) The formed array gaps between $Cu_2Se$ nanosheets can function as microreactors for controlling the liberation of reaction intermediates, as demonstrated by the in situ Fourier-transform infrared (FTIR) spectroscopy. In addition, density functional

theory (DFT) calculations suggest that charge-enriched Cu−$Cu_2Se$ interfaces can tune the adsorption and reduce the formation energy of $C_2$ intermediates. 4) The in situ grown $Cu_2O$ nanoparticles on $Cu_2Se$ nanosheets induce the formation of Z-scheme $Cu_2Se$-$Cu_2O$ (CSCO) heterojunction, which can accelerate the spatial separation of carriers along different directions (electrons to $Cu_2Se$ and holes to $Cu_2O$), thus inhibiting carrier recombination and promoting their photocatalytic efficiency. As a result, the achieved Cu-CSCO HNA system can be heated up to 200 °C within 2 min under visible-IR light irradiation and deliver an ethanol generation rate of 149.45 $\mu mol\ g^{-1}\ h^{-1}$ with an electron selectivity of 48.75% using $H_2O$ as the proton source. Our work illustrates the possibility of designing HNA system for $CO_2$ conversion to multi-carbon chemicals under solar energy.

## Results

### Theoretical screening of photothermal materials and heterostructures

In view of the fact that single photothermal materials are theoretically excluded for $CO_2$ conversion by active electron-hole pairs due to their intrinsic narrow band gap, heterostructure engineering would provide a potential approach to cope with this dilemma[18]. To construct a feasible heterostructure for simultaneous $CO_2$ reduction and $H_2O$ oxidation under light irradiation, well-matched band edge positions are the prerequisite requirement. In principle, the electron-enriched conduction band minimum (CBM) of the catalysts should be more negative than the potential for $CO_2$ reduction (e.g. 0.09 V for ethanol generation, vs. NHE at pH = 0), while the hole-enriched valance band maximum (VBM) needs to be more positive than the potential for $H_2O$ oxidation (1.23 V for $O_2$ generation, vs. NHE at pH = 0)[19]. In this context, Z-scheme heterostructures would offer a possibility to overcome the drawback of the narrow band gap in photothermal materials. Since the $Cu_2Se$ compound exhibits a superior photothermal effect, we then take it as an example for proof of concept. Given that $Cu_2O$ typically forms readily on the surface of $Cu_2Se$ compounds, we consider that the

resulting $Cu_2Se$-$Cu_2O$ composites could function as a viable heterostructure. Commencing with the computation of surface formation energies along the (001) direction with various exposed atoms, we establish the most stable slab models for subsequent density of state (DOS) and work function calculations. According to the optimized models (Supplementary Fig. 1 and Supplementary Data 1), $Cu_2Se$ possesses a band gap of 0.9 eV as discerned from the DOS (Fig. 1a), in agreement with the previous experimental results[20]. The corresponding work function is derived to be 4.75 eV from the calculated surface electrostatic potential (Fig. 1b), in which the vacuum level and Fermi level are 4.63 eV and −0.12 eV, respectively. Similarly, the band gap of $Cu_2O$ is illustrated to be 1.95 eV (Fig. 1a) (experimental band gap is around 2.0 eV)[21], and the work function is 6.13 eV inferred from the vacuum level of 4.03 eV and Fermi level of −2.10 eV (Fig. 1b). From the foregoing theoretical results, we can clearly depict the energy band structure and edge position of the formed CSCO heterostructure. As displayed in Fig. 1c, the CBM and VBM of $Cu_2Se$ are −0.65 and 0.25 V, while those for $Cu_2O$ are −0.32 and 1.63 V (vs. NHE at pH = 0), respectively. In theory, during the photoexcited processes, electrons in the valence band of $Cu_2O$ can be excited into the corresponding conduction band, followed by transfer to the valence band of $Cu_2Se$. Moreover, these valence band electrons can absorb light energy, further ascending to the high-energy conduction band of $Cu_2Se$. For photothermal materials, a fraction of these high-energy electrons normally engages with the lattice. Subsequent non-radiative transitions make these highly energetic conduction-band electrons descend to the CBM. The electron-lattice interaction thus generates heat to locally increase the temperature of catalyst[22]. Due to the existence of the band bending and built-in electric field, the active electrons in the conduction band of $Cu_2Se$ cannot be transferred back to $Cu_2O$ but participate in the reduction reaction, while the holes in the valence band of $Cu_2O$ can enable the $H_2O$ oxidation simultaneously.

## Synthesis and characterizations of Cu-CSCO HNA

Based on the above theoretical design, we then synthesized the Cu-CSCO HNA system utilizing a soak and calcination approach as displayed in Supplementary Fig. 2 (see details in Methods). By regulating the concentration of Se precursor solution and the reaction time (Supplementary Fig. 3-4), we successfully fabricated a large-area (28 × 28 cm²) $Cu_2Se$ nanosheet film on Cu foil with uniform thickness (Supplementary Fig. 5a), indicating the potential of this approach for upscalable preparation of catalyst. Scanning electron microscopy (SEM) and transmission electron microscopy (TEM) images in Supplementary Fig. 5b-d depict the smooth and flake-like morphology of the obtained $Cu_2Se$ nanosheets, whilst high-resolution TEM (HRTEM) images (Supplementary Fig. 5e) confirm their hexagonal crystalline structure along the (111) direction. The energy-dispersive spectroscopy (EDS) element mapping images reveal the even distribution of Cu and Se components (Supplementary Fig. 5 f). It is worth noting that the initially synthesized $Cu_2Se$ nanosheets demonstrate limited stability at elevated temperatures of 100 °C, so we categorize them as the low-temperature-phase $Cu_2Se$ (L-$Cu_2Se$). We found that the L-$Cu_2Se$ nanosheets can be transformed into more stable cubic high-temperature-phase $Cu_2Se$ nanosheets (H-$Cu_2Se$) by rapid calcination at 200 °C. XRD patterns (Supplementary Fig. 6) and HRTEM images (Supplementary Fig. 7) of $Cu_2Se$ nanosheets after different calcination times and temperatures clearly show the transformation process from the L-$Cu_2Se$ to the mixed phase comprising L-$Cu_2Se$ and H-$Cu_2Se$ and finally to the stable cubic H-$Cu_2Se$ nanosheets with interplanar distances of 0.285 nm. SEM and TEM images (Supplementary Fig. 8) demonstrate that the H-$Cu_2Se$ nanosheets exhibit an unchanged morphology and elemental distribution. To further adapt to the redox potential, we then constructed the CSCO heterojunction by calcinating the H-$Cu_2Se$ nanosheets in air, in which the surface of the $Cu_2Se$ compound was partially oxidized to in situ form the $Cu_2O$

nanoparticles. This dual role of $Cu_2O$ includes acting as a protective shield to ensure the stability of $Cu_2Se$, while also serving as the oxide terminus to facilitate the $H_2O$ oxidation reaction[23].

Digital images in Fig. 2a elucidate that the Cu-CSCO HNA could be readily synthesized on a scale of 28 × 28 cm². The cross-section SEM images treated by focused ion beam (FIB) milling (Fig. 2b) determine that the thickness of CSCO HNA is around 6.5 μm, which is beneficial for light absorption and would maximize the use of catalytically active sites. A large number of array gaps are observed between the vertically grown nanosheets, which could act as microreactors to increase the concentration of reaction intermediates. SEM (Fig. 2d, e) and TEM images (Fig. 2f) demonstrate that there are apparent nanoparticles on the surface of $Cu_2Se$ nanosheets while HRTEM images in Fig. 2g show the crystallinity of $Cu_2Se$ and $Cu_2O$, demonstrating their coexistence in the same nanosheet, which further confirms the formation of the CSCO heterostructure. XRD patterns result in Supplementary Fig. 9 suggest the phase change from L-$Cu_2Se$ to H-$Cu_2Se$ and finally to the CSCO heterojunctions. Besides, oxygen element can be obviously detected on the $Cu_2Se$ nanosheets according to the EDS mapping images (Fig. 2c) and X-ray photoelectron spectroscopy (XPS) spectra (Supplementary Fig. 10). Additionally, the element content analysis of H-$Cu_2Se$ and CSCO (Supplementary Fig. 11) reveal a considerable surge in oxygen concentration alongside a reduction in selenium content. All the results suggest the successful formation of CSCO NHA. It should be mentioned that our synthesis approach is suitable for various Cu substrates. We also fabricated the CSCO HNA in situ grown on Cu bulk (Supplementary Fig. 12) and Cu foam (Supplementary Fig. 13). The morphologies of the acquired CSCO HNA, whether on Cu bulk or foam substrates, closely resemble those on Cu foil.

## Study of photothermal $CO_2$ conversion

The excellent light absorption ability is the prerequisite for efficient photothermal effect and photocatalysis. UV-vis-NIR diffuse reflectance spectra in Fig. 3a show that the Cu-CSCO HNA possesses much better light absorption ability over the entire spectral range than the pure Cu foil. As displayed in Fig. 3b, the good light absorption ability endows them with a significant photothermal effect under visible-near-infrared light irradiation. Upon using different Cu substrates, we observed a variation in the intensity of the photothermal effect, as depicted in Supplementary Fig. 14. The CSCO HNA, when in situ grown on Cu foil, exhibits the most pronounced photothermal effect. This particular system is able to elevate its temperature up to approximately 200 °C in just 2 min (the corresponding illumination spectrum is shown in Supplementary Fig. 15).

To assess the catalytic performance of Cu-CSCO HNA, we performed the $CO_2$ conversion experiments under different conditions (See the Method section). For the typical process, 2 × 3 cm² as-obtained Cu-CSCO HNA was put on a hollow quartz column within the sealed quartz reactor (Supplementary Fig. 16a, b). 15 mL deionized (DI) water was added to the bottom. Xe lamp with AM 1.5 G and 400 nm cutoff filter was used as the light source. The instrument was initially evacuated three times, afterward, pumped by high-purity $CO_2$ to reach atmospheric pressure. The gas and liquid products were detected by gas chromatography (GC) and ¹H nuclear magnetic resonance (NMR) spectrum, respectively. As described in Fig. 3c, when performed under light irradiation, excluding the thermal effect by floating the catalyst film directly on the cooling water (Supplementary Fig. 16c, d), CO was detected as the main product. Conversely, $CH_4$ and $C_2H_4$ were the primary products under the pure thermal catalysis at the temperature of 200 °C (Supplementary Fig. 17). These results are well consistent with the previous reports that high temperature usually promotes the formation of C-H bonds during the $CO_2$ conversion processes[24]. In contrast, when evaluating this Cu-CSCO HNA under photothermal conditions, excellent performance for $CO_2$ reduction was obtained. As shown in Supplementary Fig. 18, GC confirms that the gas products

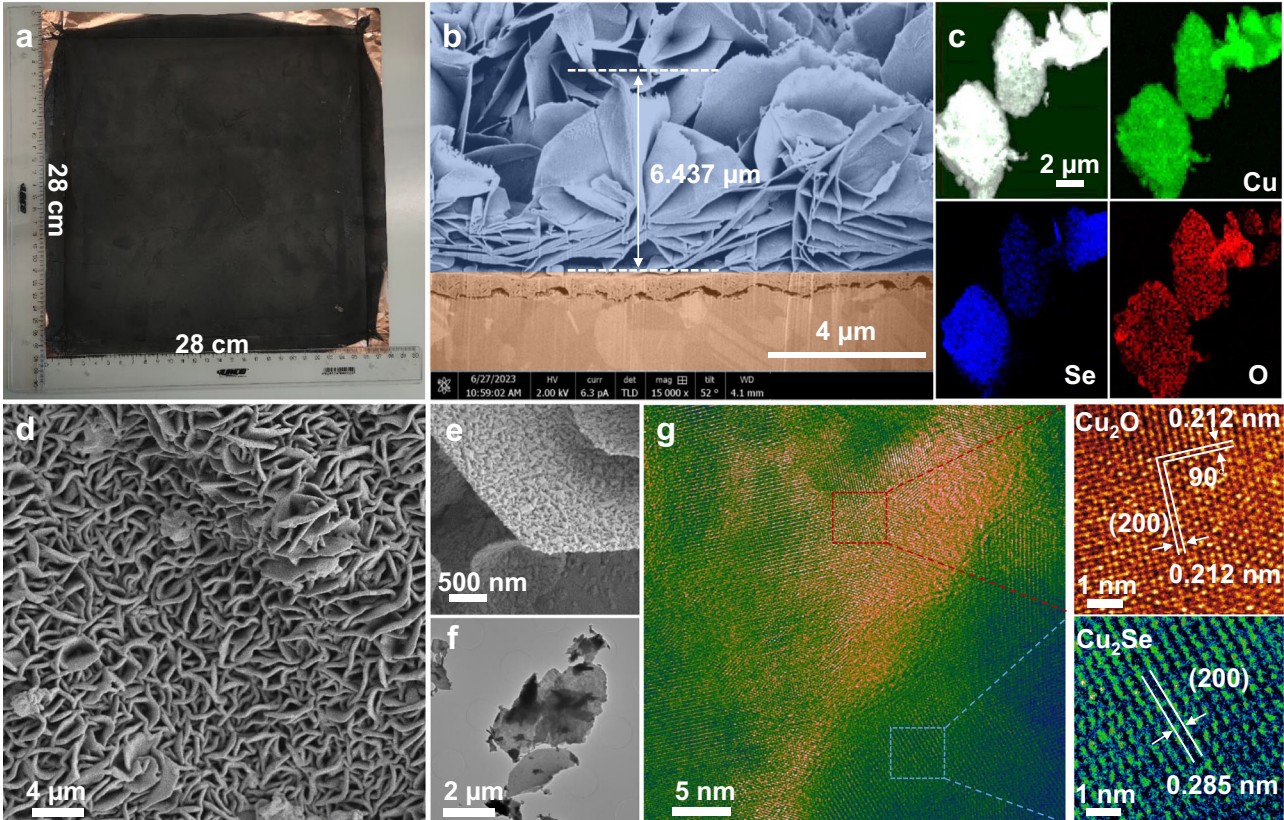

**Fig. 2 | Characterizations of Cu-CSCO HNA. a** Digital images of the large-scale (28 × 28 cm²) Cu-CSCO HNA film; (**b**) SEM images of focused ion beam (FIB) polished cross section; (**c**) STEM and EDS mapping images of each elements; (**d**, **e**) SEM images of Cu-CSCO HNA, in which nanoparticles can be clearly observed on the surface of Cu₂Se nanosheets; (**f**) TEM images of CSCO heterojunction nanosheets; (**g**) HRTEM images of CSCO heterojunction nanosheets. Source data are provided as a Source Data file.

include CO, CH₄, and C₂H₄, while O₂ is detected as the oxidation product (Supplementary Fig. 19). According to the calibration curves in Supplementary Fig. 20, the yield of CO, CH₄, and C₂H₄ is calculated to be 0.19 μmol g⁻¹ h⁻¹, 225.23 μmol g⁻¹ h⁻¹ and 6.95 μmol g⁻¹ h⁻¹, respectively (Fig. 3c and Supplementary Figs. 21, 22). It is notable that from the ¹³C and ¹H NMR spectrum in Fig. 3g and Supplementary Fig. 23, ethanol is the only liquid product generated during the CO₂ reduction, with a yield of 149.45 μmol g⁻¹ h⁻¹ and an electron selectivity of 48.75%. The corresponding apparent quantum yield (AQY) for CO₂ conversion is calculated to be 0.286%. The achieved rate of ethanol generation is nearly 3 times higher than the state-of-the-art performance in the photo and photothermal catalysis (Fig. 3d). Considering that the ethanol formation is favored under reaction conditions where methane production is also prominent[25], the excellent ethanol yield in this work can be attributed to the high concentration of CHₓ* intermediates produced during CH₄ formation. Several controlled experiments, like under N₂ atmosphere and without a catalyst, show no product during the catalysis, which elucidates that light, CO₂ reactant, and catalysts are all necessary for the effective CO₂ reduction to ethanol.

To further assess the durability of the array system during the photothermal catalysis, cycle stability tests were carried out. As illustrated in Fig. 3e, the yield for each product only shows a slight decrease after 10 cycles for 120 h. The excellent stability of the catalysts is further confirmed by the TEM, SEM, XRD, EDS, and XPS characterizations as the morphology, crystal structure, and valance state for Cu-CSCO HNA remain unchanged before and after continuous photocatalysis (Supplementary Fig. 24, 25). The isotope-labeled ¹³CO₂ mass spectrometry was performed to further unveil the source of products, in which the photon energy of 14.5 eV was selected for distinguishing the gas products of CO, CH₄, and C₂H₄ according to their absolute photoionization cross sections in Supplementary Fig. 26. As a result, only ¹³CO, ¹³CH₄ and ¹³C₂H₄ species were detected when using the isotope-labeled ¹³CO₂ as reactants (Fig. 3f), implying that the products indeed originated from photothermal CO₂ reduction. To evaluate the possibility of using Cu-CSCO HNA as the catalyst for practical application, photothermal CO₂ reduction under natural solar spectrum was carried out as shown in Fig. 3h. Even under such ambient conditions, detectable quantities of CH₄, C₂H₄, and ethanol were produced, with generation rates of 62.48, 1.44 and 53.47 μmol g⁻¹ h⁻¹, respectively (Fig. 3i and Supplementary Fig. 27). We also found that the yield of products using Cu-CSCO HNA for CO₂ photoreduction is linearly correlated to the catalyst area (Supplementary Fig. 28), suggesting that they are potentially scalable for large area applications.

## Mechanistic insights into the CO₂ photothermal reduction processes

To investigate the impact of each compound within the Cu-CSCO HNA, we also synthesized the Cu₂Se nanosheets, Cu₂O nanoparticle and CSCO heterostructure without Cu substrate for comparison. The corresponding characterizations are displayed in Supplementary Figs. 29–32. UV-vis-NIR diffuse reflectance spectra in Supplementary Fig. 33 illustrate that the formed CSCO heterostructure without Cu foil exhibits much better light absorption capacity than the pure H-Cu₂Se nanosheets and Cu₂O nanoparticle. The corresponding optical band gap of H-Cu₂Se and Cu₂O is deduced to be 0.9 eV and 1.93 eV based on the Tauc plots (Supplementary Fig. 34c, f), which are consistent with the calculated values. Besides, to unveil their band structures, ultraviolet photoelectron spectroscopy (UPS) experiments for H-Cu₂Se and Cu₂O were carried out as shown in Supplementary Figs. 34, 35. The band edge positions of H-Cu₂Se (CBM: -0.55 V; VBM: 0.35 V *vs.* NHE at

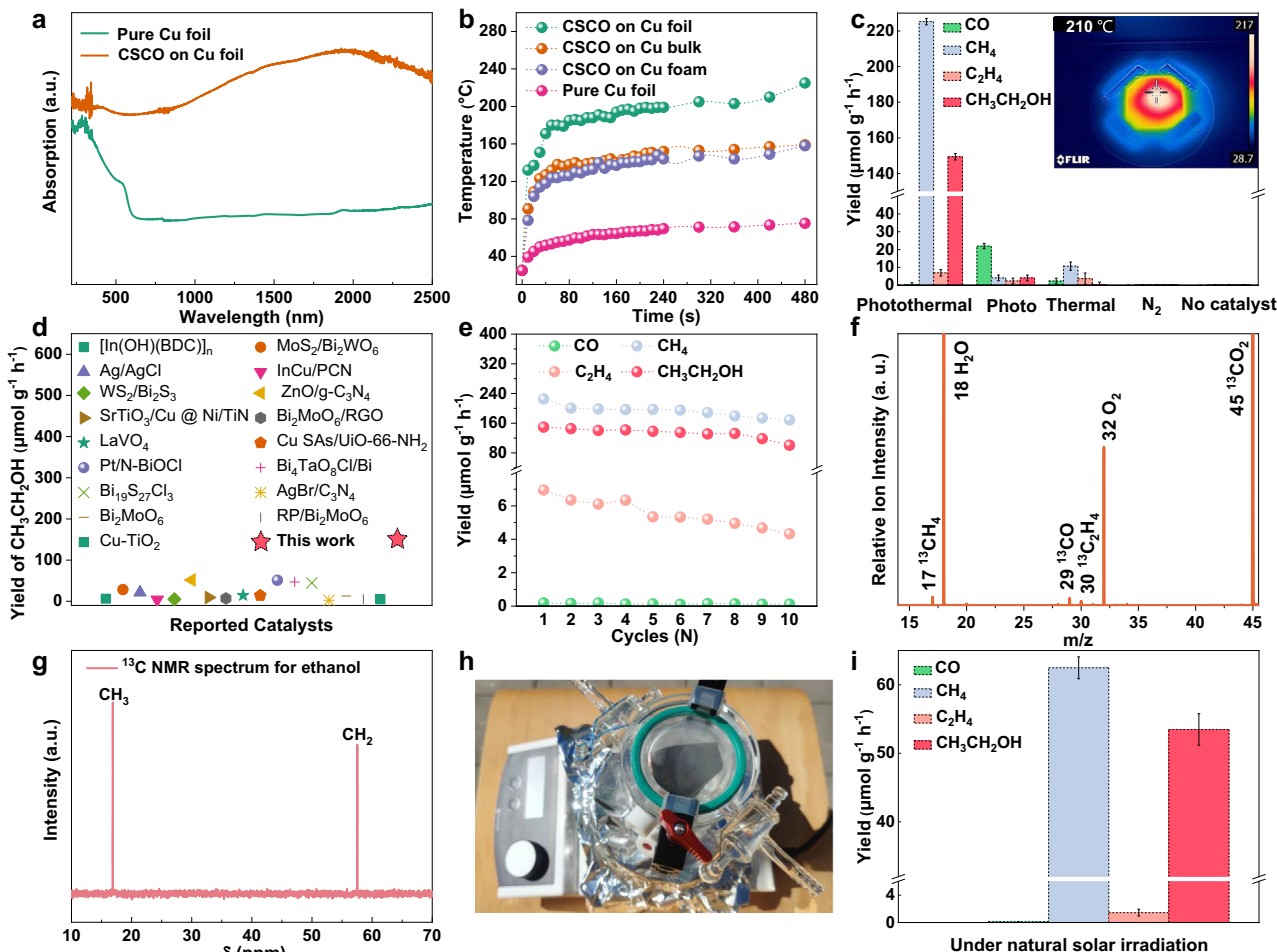

**Fig. 3 | Photothermal effect and catalytic performance of Cu-CSCO HNA.**
**a** UV–vis–NIR diffuse reflectance spectra; (**b**) photothermal effect of CSCO on different Cu substrates; (**c**) yields of photothermal $CO_2$ reduction to CO, $CH_4$, $C_2H_4$ and $CH_3CH_2OH$ over different catalysts and conditions, in which error bars represent the standard deviation (s. d.) of three independent measurements using fresh sample for each measurement, insert: thermographic photographs of Cu-CSCO HNA after 5 min irradiation under visible-near-infrared light; (**d**) the performance comparasion of ethanol in our work with that in the previous literitures, including photocatalysis and photothermal catalysis[49–65]; (**e**) cycling measurements for photothermal $CO_2$ conversion on Cu-CSCO HNA (when a new catalytic cycle begins, the reactor is pumped and refilled with pure $CO_2$), each cycle is performed for 12 h; (**f**) SVUV-PIMS spectrum of the gas products after $^{13}CO_2$ photothermal reduction on Cu-CSCO HNA at $hv = 14.5$ eV; (**g**) $^{13}C$ NMR spectrum of ethanol produced after 12 h photothermal catalysis on Cu-CSCO HNA; (**h**) digital image of photothermal $CO_2$ conversion under natural solar irradiation, where this experiment was conducted from 10 am to 4 pm on June 23–25, 2022, Dresden, Germany; (**i**) yields of photothermal $CO_2$ reduction to CO, $CH_4$, $C_2H_4$ and $CH_3CH_2OH$ over Cu-CSCO HNA under natural solar irradiation, in which error bars represent the standard deviation (s. d.) of three independent measurements using fresh sample for each measurement. Source data are provided as a Source Data file.

pH = 0) and $Cu_2O$ (CBM: -0.31 V; VBM: 1.62 V *vs.* NHE at pH = 0) are in agreement with the theoretical results in Fig. 1e. XPS and photoluminescence (PL) spectra were further executed to determine the electron transfer in the CSCO heterostructure. As illustrated in Supplementary Fig. 36a, b, the characteristic peaks of Se $3d$ in CSCO heterostructure exhibit a noticeable red shift of 0.72 eV compared to that in the pure H-$Cu_2Se$ nanosheets. Conversely, the corresponding O $1s$ peaks in CSCO heterostructure shift towards a high-energy direction by 0.77 eV compared to that in the pure $Cu_2O$ nanoparticle. This result validates the electron transfer from $Cu_2O$ to $Cu_2Se$ within the CSCO heterostructure, indicating the formation of the predicted Z-scheme heterostructure after contacting these two compounds. PL spectra further show that the CSCO possesses the best performance for charge carrier separation with the lowest PL signal compared with the pure H-$Cu_2Se$ and $Cu_2O$ (Supplementary Fig. 36c).

To reveal the role of Cu substrate in the catalysis, the CSCO heterostructure powder sprayed on the quartz (CSCO-Q) and copper foil (CSCO-C) is respectively used as the catalyst for $CO_2$ photoreduction. As shown in Fig. 4a and Supplementary Fig. 37, upon using CSCO-Q, CO

is detected as the main product with yield of 77.86 μmol $g^{-1}$ $h^{-1}$, accompanied by the $CH_4$ generation of 40.57 μmol $g^{-1}$ $h^{-1}$. This demonstrates the ability of CSCO heterostructure without Cu substrate for photocatalytic $CO_2$ conversion. One can clearly see that the CSCO-C shows activity for CO (41.76 μmol $g^{-1}$ $h^{-1}$) and $CH_4$ (57.57 μmol $g^{-1}$ $h^{-1}$) generation (Supplementary Fig. 38). The higher $CH_4$ performance in the Cu foil system compared to the quartz system is induced by the local thermal effect of infrared insulating Cu substrate (Supplementary Fig. 39). The mechanically mixed $Cu_2Se$ nanosheet and commercial $Cu_2O$ powder exhibits a poor activity (1.24 μmol $g^{-1}$ $h^{-1}$ for CO and 0.68 μmol $g^{-1}$ $h^{-1}$ for $CH_4$) as shown in Supplementary Fig. 40, which confirms that the in situ growing of $Cu_2O$ on the $Cu_2Se$ nanosheet can ensure the close contact between these two compounds and forming the Z-scheme heterostructure, which can promote the separation and transport of the excited charge carriers, further improving its performance.

To further explore the effect of the nanosheet array configuration on the performance of $CO_2$ photoreduction, we compared the performance of CSCO-C and Cu-CSCO HNA without gap

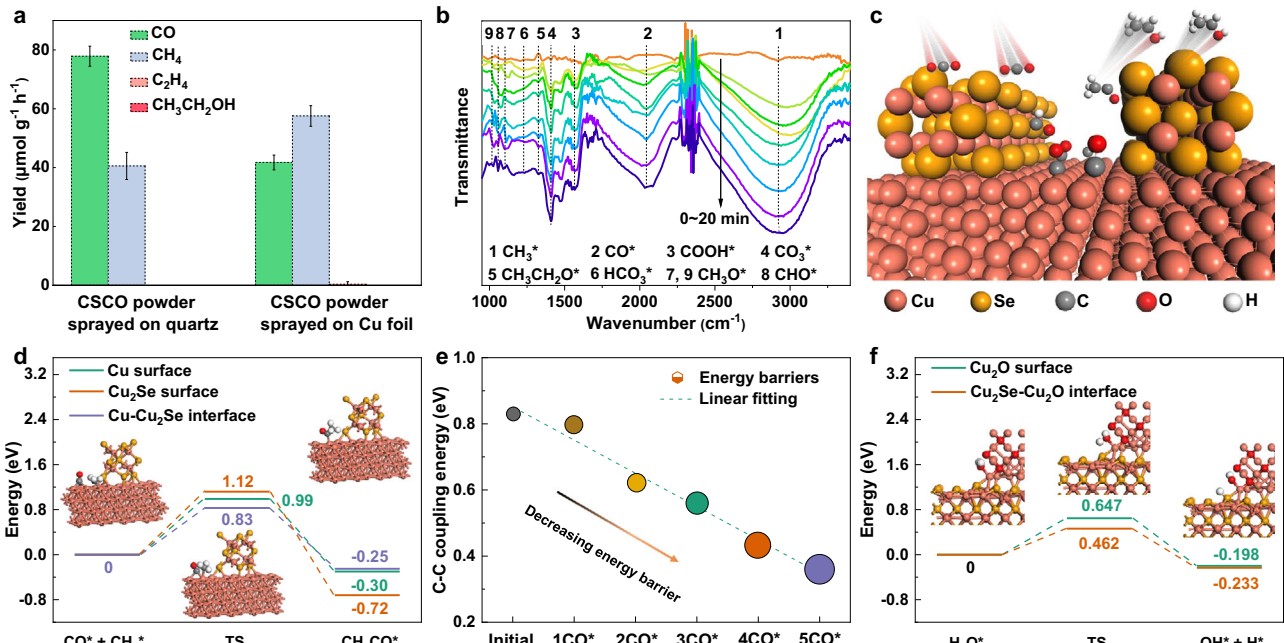

**Fig. 4 | Reaction mechanism investigation of Cu-CSCO HNA for photothermal CO₂ conversion to ethanol. a** Performance of CSCO powder sprayed on quartz and Cu foil; (**b**) in situ FTIR spectroscopy characterization for co-adsorption of a mixture of CO₂ and H₂O vapor under light irradiation over in situ grown Cu-CSCO HNA; (**c**) schematic illustration of array gaps, which enrich the key intermediates to accelerate ethanol generation from CO₂ conversion on Cu–Cu₂Se interfaces; (**d**) C–C coupling TS energy barriers of CO* and CH₃* intermediates on different surfaces, the insert structure images are the initial, TS and final configuration of the intermediates at Cu–Cu₂Se interface during the coupling process; (**e**) C–C coupling TS energy barriers of CO* and CH₃* intermediates at Cu–Cu₂Se interface with different CO* coverage; (**f**) TS energy barriers of H₂O* dissociation into OH* and H* on the pure Cu₂O surface and Cu₂Se–Cu₂O interface, the insert structure images are the initial, TS and final configuration of the intermediates at Cu₂Se–Cu₂O interface during the dissociation process. Source data are provided as a Source Data file.

(Supplementary Fig. 41) under the same conditions with that used for in situ grown Cu-CSCO HNA with gap. It is notable that there is only trace C₂H₄ and almost no liquid product produced over the CSCO-C system (Supplementary Fig. 38) while small amounts of C₂H₄ and ethanol were detected over Cu-CSCO HNA without gap (Supplementary Fig. 42). The poor activity of the sprayed system and Cu-CSCO HNA without gap for C₂₊ products can be attributed to the less concentration of C₁ reaction intermediates like CO* and CH₃*, which can reduce the possibility of C–C coupling and limit the generation of C₂₊ products[26]. A distinction between the CSCO-C and Cu-CSCO HNA without gap with pure Cu-CSCO HNA can be attributed to the numerous spatial gaps between the vertically arranged Cu₂Se nanosheets within the in situ grown Cu-CSCO HNA (Supplementary Figs. 41–43), which provides favorable conditions for the adsorption of C₁ intermediates as illustrated in Fig. 4c. To support the proposed facilitation mechanism of the array gaps and Cu–Cu₂Se interfaces, DFT calculations are performed based on the constructed theoretical models (Supplementary Figs. 44–45 and Supplementary Data 1). The charge density difference (CDD) in Supplementary Fig. 46 shows that the charges are enriched at the Cu–Cu₂Se interface, which is beneficial for the intermediate adsorption. This result is confirmed by the calculated adsorption energy for CO species in Supplementary Fig. 47, from which a stronger CO adsorption energy of −0.89 eV is obtained at the Cu-Cu₂Se interface than that on the pure Cu₂Se surface (−0.68 eV). Finite-element method simulations (Supplementary Fig. 48) and CO₂ adsorption isotherms (Supplementary Fig. 49) further confirmed that the microreactors induced by Cu₂Se nanosheet array gaps can not only enhance the CO₂ adsorption on the catalyst surface but also increase the local concentration of CH₃* and CO* intermediates. This localization effect is well-consistent with the CO temperature-programmed desorption (TPD) measurements (Supplementary Fig. 50), in which the temperature of CO desorption for

the Cu-CSCO HNA without gap (treated by mechanical friction method) is 136 °C, much smaller than that for the Cu-CSCO HNA with gap (213 °C), confirming the CO desorption is harder in the Cu-CSCO HNA system with gap. It is worth noting that the Cu–Cu₂Se interface possesses a good adsorption capacity but exhibits a higher energy barrier for C₁ intermediate (CO*, CHO*, CHₓ*) generation. As displayed in Supplementary Fig. 51, the pure Cu₂Se surface is more favorable to produce CO and CHO* species with lower Gibbs free energy barrier of 0.15 and 0.51 eV, respectively (in comparison to 0.40 and 1.04 eV at the Cu–Cu₂Se interface). Therefore, numerous C₁ intermediates can be produced on the Cu₂Se surface and subsequently transferred to the Cu–Cu₂Se interface.

To gain insight into the possible reaction mechanism, in situ FTIR spectrum is performed during the CO₂ photoreduction processes over in situ grown Cu-CSCO HNA system and the sprayed CSCO-C. As shown in Fig. 4b and Supplementary Fig. 52, a series of new infrared peaks at around 1568 cm⁻¹ are detected for both samples, which are attributed to the COOH* group, a crucial initial intermediate for CO₂ reduction[27,28]. The absorption bands near 1106 and 1025 cm⁻¹ belong to the CH₃O* group, and the peaks at 1061 cm⁻¹ are assigned to the characteristic bands of CHO*; both the CH₃O* and CHO* groups are pivotal intermediates of CO₂ photoreduction to hydrocarbon[29,30]. Moreover, the peaks at 2042 and 2920 cm⁻¹ can be attributed to the adsorbed CO* and CH₃* species, respectively[31–33]. And the peak at 1337 cm⁻¹ belongs to the adsorbed CH₃CH₂O*[34,35]. It is notable that the relative intensities of the aforementioned three peaks (CO*, CH₃* and CH₃CH₂O*) are higher within the Cu-CSCO HNA system, suggesting the higher concentration of the corresponding reaction intermediates. Besides, the peaks at 1218 cm⁻¹ are inferred to the asymmetric stretching of HCO₃* group, while the peaks at 1410 cm⁻¹ are indexed to the CO₃* group[27,36]. Besides, in situ Raman spectra were also conducted to confirm the reaction mechanisms as displayed in Supplementary

Fig. 53. As suggested by the detected reaction intermediates, the potential reaction pathway can be inferred as follows:

$$* + CO_2 + e^- + H^+ \rightarrow COOH^* \tag{1}$$

$$COOH^* + e^- + H^+ \rightarrow CO^* + H_2O \tag{2}$$

$$CO^* + e^- + H_+ \rightarrow CHO^* \text{ or } CO^* \rightarrow CO \uparrow + * \tag{3}$$

$$CHO^* + e^- + H^+ \rightarrow CH_2O^* \tag{4}$$

$$CH_2O^* + e^- + H^+ \rightarrow CH_3O^* \tag{5}$$

$$CH_3O^* + e^- + H^+ \rightarrow CH_4 \uparrow + O^* \text{ or } CH_3O^* + e^- + H^+ \rightarrow CH_3OH^* \tag{6}$$

$$CH_3OH^* + e^- + H^+ \rightarrow CH_3^* + H_2O \tag{7}$$

$$CH_3^* + CO^* \rightarrow CH_3CO^* \tag{8}$$

$$CH_3CO^* + 3(e^- + H^+) \rightarrow CH_3CH_2OH + * \tag{9}$$

The kinetic effects, governed by transition barriers along the pathway, also control the reaction selectivity. Based on the climbing image-nudged elastic band (CI-NEB) method, we carried out the transition state (TS) simulations of C–C coupling processes at pure Cu surface (Supplementary Fig. 54), pure $Cu_2Se$ surface (Supplementary Fig. 55) and Cu–$Cu_2Se$ interface (Supplementary Fig. 56). The C–C coupling energy barriers of CO* and CH$_3$* intermediates are treated as the criterion to determine the activity for ethanol generation[37,38]. As shown in Fig. 4d, Cu–$Cu_2Se$ interfaces render a lower TS energy barrier of 0.83 eV for C–C coupling compared to that on pure Cu surface (0.99 eV) and $Cu_2Se$ surface (1.12 eV), indicating their higher activity for $CO_2$ conversion to ethanol. In addition, to investigate the effect of local concentration of C$_1$ intermediates on C–C coupling processes, TS energy barriers of CO* and CH$_3$* at Cu–$Cu_2Se$ interfaces with different CO coverage are conducted (illustrated in Fig. 4e). With increasing the CO coverage, the TS energy barriers of C–C coupling gradually decrease from 0.83 eV to 0.36 eV, suggesting that the locally higher concentration of CO* intermediates can obviously improve C–C coupling processes, thus enhancing the performance for ethanol generation (Supplementary Figs. 57–61).

It is noted that $CO_2$ reduction reaction requires the participation of protons (H*), especially for producing C-H compounds (methane, ethylene, and ethanol). As the sole source of H*, the $H_2O$ dissociation process ($H_2O^* \rightarrow H^* + OH^*$) plays a vital role in $CO_2$ protonation but is often overlooked in previous studies. Here, we also investigated the $H_2O$ dissociation processes at the pure $Cu_2O$ surface and CSCO interface. The structures that have been optimized by exploring the global minimum energy along the x, y, and z directions (Supplementary Fig. 62) reveal an elevated charge density at the CSCO interface (Supplementary Fig. 63). These optimized structures are subsequently employed to calculate the TS energy barriers. As displayed in Fig. 4f and Supplementary Fig. 64–65, the energy barrier of $H_2O$ dissociation at the CSCO interface (0.462 eV) is significantly lower than that at the pure $Cu_2O$ surface (0.647 eV).

## Discussion

In summary, we demonstrated a large-area Cu-CSCO HNA photothermal catalyst for achieving highly efficient $CO_2$ conversion into ethanol under the solar spectrum. Our designed catalysts take advantage of the synergistic photo-thermal effect strategy, spatially separated active sites, and array gap-induced microreactor, which can promote carrier transmission and tune the local intermediate concentration, thereby increasing the C–C coupling probability. As a result, the Cu-CSCO HNA catalyst achieved an ethanol yield of 149.45 $\mu mol\ g^{-1}\ h^{-1}$, nearly 3 times higher than the state-of-the-art performance in the photo and photothermal catalysis (<50 $\mu mol\ g^{-1}\ h^{-1}$). The design of this photothermal catalyst system not only holds great potential for applications in large-scale production of high value-added multi-carbon compounds and the energy storage in the form of chemical fuels, but also exhibits substantial promise for various catalytic reactions, particularly those demanding elevated temperatures, such as CH$_4$ oxidation and NH$_3$ synthesis.

## Methods
### Materials
Cu foil (99.98%), Cu powder (99.999% trace metals basis), $Cu_2O$ powder (≥99.99% trace metals basis, anhydrous), NaOH (pellets for analysis EMSURE®), Se powder (99.99% trace metals basis) and NaBH$_4$ (99.99% trace metals basis) are all acquired from Sigma-Aldrich and were used without any further purification. DI water with resistivity of 18.2 MΩ.cm is obtained by the ultra-pure water system from Stakpure GmbH, Germany.

### Catalysts synthesis
**Synthesis of in situ grown L-Cu$_2$Se nanosheet on Cu foil.** 4 g NaOH was added into 25 mL DI water and sonicating the solution until the solids are completely dissolved. Afterwards, 40 mg Se powder and 189 mg NaBH$_4$ were sequentially added to the above solution. After sonication and shaking for 10 min, an orange-red transparent solution was obtained. Subsequently, 10 mL above solution was slowly dropped into the 3 × 3 cm$^2$ Cu foil. After standing for additional 2 h, washing the surface of Cu foil with DI water for 4 times. The desired sample can be obtained after vacuum drying at 60 °C for 5 h.

**Synthesis of in situ grown H-Cu$_2$Se nanosheet on Cu foil.** In a typical procedure, the as-obtained sample of L-Cu$_2$Se nanosheet on Cu foil was calcined at 200 °C with a heating rate of 10 °C min$^{-1}$ for 45 min under the Ar atmosphere and then cooled to room temperature. The obtained sample, called as H-Cu$_2$Se nanosheet on Cu foil, was taken out for further characterization.

**Synthesis of in situ grown Cu-CSCO HNA on Cu foil.** The preparation procedure is the same with that for H-Cu$_2$Se nanosheet on Cu foil, the only difference is that the calcination atmosphere is in the air. 4 g NaOH was added into 25 mL DI water and sonicating the solution until the solids are completely dissolved. Afterwards, 40 mg Se powder and 189 mg NaBH$_4$ were sequentially added to the above solution. After sonication and shaking for 10 min, an orange-red transparent solution was obtained. Subsequently, 10 mL above solution was slowly dropped into the 3 × 3 cm$^2$ Cu foil. After standing for additional 2 h, washing the surface of Cu foil with DI water for 4 times and vacuum drying it at 60 °C for 5 h. The as-obtained sample of L-Cu$_2$Se nanosheet on Cu foil was calcined at 200 °C with a heating rate of 10 °C min$^{-1}$ for 45 min under the air atmosphere and then cooled to room temperature. The obtained Cu-CSCO HNA on Cu foil was taken out for further characterization. The mass loading of CSCO heterojunction-nanosheet arrays is calculated to 1.52 mg cm$^{-1}$ by comparing the mass difference of Cu foil before and after synthesis processes.

**Synthesis of in situ grown Cu-CSCO HNA on Cu bulk.** The preparation process is the same as above, the only difference is that the substrate is changed to Cu bulk. 4 g NaOH was added into 25 mL DI water and sonicating the solution until the solids are completely dissolved. Afterwards, 40 mg Se powder and 189 mg NaBH$_4$ were sequentially added to the above solution. After sonication and shaking for 10 min,

an orange-red transparent solution was obtained. Subsequently, 10 mL above solution was slowly dropped into the $3 \times 3$ cm$^2$ Cu bulk. After standing for additional 2 h, washing the surface of Cu foil with DI water for 4 times and vacuum drying it at 60 °C for 5 h. The as-obtained sample of L-Cu$_2$Se nanosheet on Cu bulk was calcined at 200 °C with a heating rate of 10 °C min$^{-1}$ for 45 min under the air atmosphere and then cooled to room temperature. The obtained Cu-CSCO HNA on Cu bulk was taken out for further characterization.

**Synthesis of in situ grown Cu-CSCO HNA on Cu foam.** The preparation process is the same as above, the only difference is that the substrate is changed to Cu foam. 4 g NaOH was added into 25 mL DI water and sonicating the solution until the solids are completely dissolved. Afterwards, 40 mg Se powder and 189 mg NaBH$_4$ were sequentially added to the above solution. After sonication and shaking for 10 min, an orange-red transparent solution was obtained. Subsequently, 10 mL above solution was slowly dropped into the $3 \times 3$ cm$^2$ Cu foam. After standing for additional 2 h, washing the surface of Cu foil with DI water for 4 times and vacuum drying it at 60 °C for 5 h. The as-obtained sample of L-Cu$_2$Se nanosheet on Cu foam was calcined at 200 °C with a heating rate of 10 °C min$^{-1}$ for 45 min under the air atmosphere and then cooled to room temperature. The obtained Cu-CSCO HNA on Cu foam was taken out for further characterization.

**Synthesis of independent L-Cu$_2$Se nanosheet powder without Cu substrate.** 4 g NaOH was added into 25 mL DI water and sonicating the solution until the solids are completely dissolved. Afterwards, 40 mg Se powder and 189 mg NaBH$_4$ were sequentially added to the above solution. After sonication and shaking for 10 min, an orange-red transparent solution was obtained. Subsequently, 55 mg Cu powder was added into the above solution. After stirring for additional 4 h, the final product was collected by centrifuging the mixture, washed with ethanol and DI water for several times until the unreacted residuals were completely removed, and then dried in vacuum oven at 60 °C for 5 h. The black powder was obtained for further usage.

**Synthesis of independent H-Cu$_2$Se nanosheet powder without Cu substrate.** In a typical procedure, 50 mg of the as-obtained sample of L-Cu$_2$Se nanosheet powder was calcined at 200 °C with a heating rate of 10 °C min$^{-1}$ for 45 min under the Ar atmosphere and then cooled to room temperature. The obtained sample, called as H-Cu$_2$Se nanosheet powder, was taken out for further characterization.

**Synthesis of Cu$_2$Se-Cu$_2$O heterojunction-nanosheet powder without Cu substrate.** The preparation procedure is the same as that for H-Cu$_2$Se nanosheet powder, the only difference is that the calcination atmosphere is in the air. 50 mg the as-obtained sample of L-Cu$_2$Se nanosheet powder was calcined at 200 °C with a heating rate of 10 °C min$^{-1}$ for 45 min under the air atmosphere and then cooled to room temperature. The obtained sample, called as Cu$_2$Se-Cu$_2$O heterojunction-nanosheet powder, was taken out for further characterization.

**Characterization**
X-ray photoelectron spectroscopy (XPS) spectra were acquired on an ESCALAB MKII system with Al Kα ($hv = 1486.6$ eV) as the excitation source. The binding energies obtained in the XPS spectral analysis were corrected for specimen charging by referencing C 1s to 284.8 eV. TEM and HRTEM images were performed with a JEOL Jem F-200C TEM with an acceleration voltage of 200 kV. XRD patterns were obtained from a Philips X'Pert Pro Super diffractometer with Cu Kα radiation ($\lambda = 1.54178$ Å). In-situ FTIR spectra were obtained by using a Thermo Scientific Nicolet iS50. UV−vis−NIR diffuse reflectance spectra were measured on a Perkin Elmer Lambda 950 UV-vis-NIR spectrophotometer. Synchrotron-radiation photoemission spectroscopy

(SRPES) was executed at the National Synchrotron Radiation Laboratory (NSRL) in Hefei, China. Ultraviolet photoelectron spectroscopy (UPS) was performed at the Catalysis and Surface Science Endstation at the BL11U beamline of the National Synchrotron Radiation Laboratory (NSRL). The workfunction (WF) was determined by the difference between the photon energy and the binding energy of the secondary cutoff edge. To be exact, $E_B = hv - (E_K + 4.3 - 5.0)$ and WF $= hv - (E_{cutoff} - E_F)$ ($E_B$, binding energy; $hv$, photon energy; $E_K$, kinetic energy; $E_{cutoff}$, secondary cutoff edge; $E_F$, Fermi level; photon energy of 40.0 eV and a sample bias of -5 V applied to observe the secondary electron cutoff). TPD of the samples was performed using a Micromeritics ChemiSorb 2720 with a thermal conductivity detector. Fluorescence emission decay spectra were recorded with a DeltaFlex-NL (HORIBA Scientific) spectrometer. CO$_2$ adsorption isotherms measurements for all the synthetic samples were carried out using an automatic microporous physical and chemical gas adsorption analyser (ASAP 2020 M PLUS)[39]. In situ Raman spectra were recorded on confocal microscopic LabRamHR Evolution System with twice scan times. Note that when collecting the spectrum, it is necessary to briefly block the light from the Xenon lamp for a short time, in order to avoid the interference of the Xenon lamp illumination on the detection laser.

**In situ FTIR spectra experiments**
All FTIR spectra were recorded on Thermo Scientific Nicolet iS50. The spectra were displayed in transmission units and acquired with a resolution of 4 cm$^{-1}$, using 64 scans. The dome of the reaction cell had two KBr windows allowing IR transmission and a third window allowing transmission of irradiation introduced through a liquid light guide that connects to the same IR-light lamp. The catalysts were first added to the reaction cell and then trace amounts of water were sprayed on the surface of catalysts. After degassed in N$_2$ atmosphere for 20 min, the gas flow was switched to high-purity CO$_2$ until the adsorption is saturated, then the reaction cell was sealed. Next, the FTIR spectra were recorded as a function of time to investigate the dynamics of the reactant adsorption in the dark and desorption/conversion under irradiation[39].

**Photothermal CO$_2$ reduction tests under visible−near−infrared light irradiation**
For CO$_2$ photoreduction tests, we used the same method as in the previous literature[39]. For powder catalysts, before performing the CO$_2$ photoreduction performance, we fabricated the sample into a thin film: the sample was dispersed in deionized water to gain a concentration of about 5.5 mg mL$^{-1}$, and then, through spin-dropping 2 mL of the above dispersion on a quartz glass or Cu foil, followed by heat treatment at 65 °C for 30 min, the catalysts thin film could be achieved. For Cu-CSCO HNA on Cu foil, $2 \times 3$ cm$^2$ as-obtained sample was used for catalysis. During the CO$_2$ photothermal process, a MC-PF-300C Xe lamp with AREF (full spectrum reflectance 200−2500 nm), AM 1.5 G filter and 400 nm cutoff filter was used to simulate visible-near-infrared light, the corresponding illumination spectrum of which in comparison with sunlight is displayed in Supplementary Fig. 15. Note that the distance from the lamp to the sample was ∼10 cm, and the irradiation area of sample is around 9 cm$^2$ with an output light density of ∼85 mW cm$^{-2}$. The instrument was initially evacuated three times, afterwards, pumped by high-purity CO$_2$ to reach atmospheric pressure. For the catalysis excluding the heating effect, the Cu-CSCO HNA on Cu foil floated on 50 mL of water with the homothermal condensate water, which could enable the catalysts to retain a constant temperature of $290 \pm 0.2$ K. For the thermocatalysis without light, temperature controlled thermal reactor is used during the same catalytic process. The gas products were quantified by the Agilent GC-8860 gas chromatograph equipped with TDX-01 column, thermal conductivity detector (TCD) and flame ionization detector (FID) while ultrahigh-purity argon was used as a carrier gas (FID detector for carbon-based

products and TCD detector for $H_2$). The liquid products were quantified by nuclear magnetic resonance (NMR) (Bruker AVANCE AV III 400) spectroscopy, in which dimethyl sulfoxide (DMSO, Sigma, 99.99%) was used as the internal standard. The specific operation methods are as follows: We first took 400 μL solution from the bottom of the reactor after the catalysis and mixed it with 100 μL deuterated water ($D_2O$). Then 20 μL dimethyl sulfoxide (DMSO) (diluted 10,000 times with the concentration of 1.413 mM) was added as internal standards for the $^1$H NMR analysis. The area of product peaks was compared to the area of DMSO (at a chemical shift of 2.6 ppm). In our case, the triple peak of ethanol at a chemical shift of 1.06 ppm was used to calculate the generation rate (N) as the following equation:

$$N = \frac{S \times V3 \times n \times 6 \times V2}{3 \times V1 \times t \times m} \quad (10)$$

where S is the area of triple peak of ethanol compared to DMSO (identified to 1 as reference), V3 is the volume of water (15 mL) used during photothermal catalysis, n is the concentration of diluted DMSO (1.413 mM), V2 is the volume of DMSO (20 μL), V1 is the volume of the reaction solution tested (400 μL), t is reaction time and m is the mass of catalyst.

The product selectivity for $CO_2$ reduction to ethane and ethanol has been calculated using the following equation:

$$\text{Product selectivity of } C_2H_4(\%) = [n(C_2H_4)]/[n(CO) + n(CH_4) \\ + n(C_2H_4) + n(CH_3CH_2OH)] \times 100\% \quad (11)$$

$$\text{Product selectivity of } CH_3CH_2OH(\%) = [n(CH_3CH_2OH)]/[n(CO) + n(CH_4) \\ + n(C_2H_4) + n(CH_3CH_2OH)] \times 100\% \quad (12)$$

The electron selectivity for $CO_2$ reduction to $C_2H_4$ and $CH_3CH_2OH$ ($12e^-$ for the formation of $C_2H_4$ or $CH_3CH_2OH$) has been calculated using the following equation:

$$\text{Electron selectivity of } C_2H_4(\%) = [12n(C_2H_4)]/[2n(CO) + 8n(CH_4) + 12n(C_2H_4) \\ + 12n(CH_3CH_2OH)] \times 100\% \quad (13)$$

$$\text{Electron selectivity of } CH_3CH_2OH(\%) = [12n(CH_3CH_2OH)]/[2n(CO) + 8n(CH_4) \\ + 12n(C_2H_4) + 12n(CH_3CH_2OH)] \times 100\% \quad (14)$$

where n(CO), n($CH_4$), n($C_2H_4$) and n($CH_3CH_2OH$) are the amounts of produced CO, $CH_4$, $C_2H_4$ and $CH_3CH_2OH$.

## Apparent quantum yield (AQY)

The AQY is defined by the ratio of the effective electrons used for $CO_2$ conversion to the total input photon flux[40,41].

$$AQY\% = \frac{\text{Effective electrons}}{\text{Total photons}} \times 100\% = \frac{e(n) \times Y(n) \times N}{\Theta \times T \times S} \times 100\% \quad (15)$$

$$\Theta = \frac{I}{S \times T} \quad (16)$$

$$I = \frac{P \times T \times \bar{\lambda}}{h \times c} \quad (17)$$

$$P = \bar{E} \times S \quad (18)$$

$$\bar{\lambda} = \frac{\int_{\Delta\lambda} \lambda E(\lambda)d\lambda}{E} \quad (19)$$

where Y(n) is the yield of different products, including carbon monoxide, methane, ethylene and ethanol, e(n) is the required electron number for each product, N is Avogadro's number, T is the irradiation time, Θ is the photon flux, S is the illumination area, I is the incident photon number, h corresponds to the Planck constant, c stands for the speed of light, λ refers to the wavelength, $\bar{\lambda}$ refers to the average wavelength, $\bar{E}$ is the average optical power density, E is total radiation intensity, and E(λ) is the spectrum radiation intensity.

The following calculation example is based on the data from $CO_2$ photoreduction with Cu-CSCO HNA for 12 h: Y(CO, $CH_4$, $C_2H_4$, $C_2H_5OH$) = 3.12 × $10^{-8}$ mol, 3.70 × $10^{-5}$ mol, 1.14 × $10^{-6}$ mol, 2.45 × $10^{-5}$ mol; e(CO, $CH_4$, $C_2H_4$, $C_2H_5OH$) = 2, 8, 12, 12; N = 6.022 × $10^{23}$ mol$^{-1}$; T = 12 h, S = 9 cm$^2$; integration photons 400–2500 nm, Θ = 3.27 × $10^{17}$ s$^{-1}$ cm$^{-2}$. For Cu-CSCO HNA: AQY% = [(2 × 3.12 × $10^{-8}$ + 8 × 3.70 × $10^{-5}$ + 12 × 1.14 × $10^{-6}$ + 12 × 2.45 × $10^{-5}$) × 6.022 × $10^{23}$]/(3.27 × $10^{17}$ × 12 × 3600 × 9) = 0.286%.

## DFT calculation details

Density functional theory (DFT) calculations were carried out on a Vienna Ab initio Simulation Package (VASP)[42]. The exchange-correlation potential was described by the generalized gradient approximation (GGA) within the framework of Perdew-Burke-Ernzerhof (PBE) functional[43]. DFT-D3 method was employed to calculate the van der Waals (vdW) interaction[44]. The parameters of dipole correction were applied for the calculation of slab models. Electronic energies were computed with the tolerance of $1 \times 10^{-4}$ eV and a total force of 0.01 eV/Å. A kinetic cutoff energy of 450 eV was adopted. The crystal lattice parameters of $Cu_2Se$ bulk are as follows: a = b = c = 5.8358 Å (α = β = γ = 90°); while the corresponding parameters for $Cu_2O$ are a = b = c = 4.2920 Å (α = β = γ = 90°). A Monkhorst−Pack k-mesh of 8 × 8 × 8 and 6 × 6 × 6 k-points were used in the structural relaxation for $Cu_2O$ and $Cu_2Se$ bulk, respectively. The $Cu_2O$ and $Cu_2Se$ slabs were modeled by the corresponding exposed surface along (001) direction with the thickness of two unit cells, in which half bottom atoms are fixed to simulate the bulk structure. A Monkhorst−Pack k-mesh of 3 × 3 × 1 and 5 × 5 × 1 k-points were used in the structural relaxation for $Cu_2O$ and $Cu_2Se$ slabs, respectively. A vacuum space of 15 Å was inserted in the z direction to avoid interactions between periodic images. For Cu−$Cu_2Se$ heterostructures, Cu (111) slab is used to simulate the Cu foil substrate. The initial structure is obtained after computing their single point energy of heterostructures with $Cu_2Se$ slabs shifting along the x, y, and z directions. A Monkhorst−Pack k-mesh of 1 × 5 × 1 k-points was used in the structural relaxation. For $Cu_2Se$-$Cu_2O$ heterostructures, 4 × 4 $Cu_2Se$ supercell is built to hold the $Cu_2O$ cluster. The initial structure is obtained after computing their single point energy of heterostructures with $Cu_2O$ cluster shifting along the x, y, and z directions. A single Gamma k-points was used in the structural relaxation. All the information of the optimized structures is provided as Supplementary Data 1.

The surface energy $E_s$ is the energy required to cleave a surface from the corresponding bulk crystal. It can be given by

$$E_s = 1/2A[E_s(\text{unrelax}) - N \times E_b] + 1/A[E_s(\text{relax}) - E_s(\text{unrelax})] \quad (20)$$

where A is the area of the surface on the slab models, $E_s$(unrelax) and $E_s$(unrelax) represent the energy of the unrelaxed and relaxed surface slab models, respectively. N is the number of in the slab and $E_b$ is the energy of each atom in the bulk counterpart.

Adsorption energies $E_{adsorption}$ are given with reference to the isolated surface $E_{surface}$ relaxed upon removing the molecule from the unit cell using identical computational parameters and the energy of

the molecule $E_{molecule}$.

$$E_{adsorption} = E_{molecule\ on\ surface} - E_{surface} - E_{molecule} \qquad (21)$$

The computational hydrogen electrode (CHE)[45] model was used to calculate the Gibbs free energy change (ΔG) of $CO_2$ reduction reaction steps:

$$G = E_{DFT} + E_{ZPE} - TS \qquad (22)$$

$$E_{ZPE} = \sum_i 1/2 h\nu_i \qquad (23)$$

$$\Theta_i = h\nu_i/k \qquad (24)$$

$$S = \sum_i R \left[ \ln(1 - e^{-\Theta i/T})^{-1} + \Theta_i/T(e^{\Theta i/T} - 1)^{-1} \right] \qquad (25)$$

where $E_{DFT}$ is the electronic energy calculated for specified geometrical structures, $E_{ZPE}$ is the zero-point energy, $S$ is the entropy, $h$ is the Planck constant, $\nu$ is the computed vibrational frequencies, $\Theta$ is the characteristic temperature of vibration, $k$ is the Boltzmann constant, and $R$ is the molar gas constant. For adsorbates, all 3N degrees of freedom were treated as frustrated harmonic vibrations with negligible contributions from the catalysts' surfaces.

The climbing image nudged elastic band (CI-NEB) method is used to evaluate the energy barriers of transition states during C–C coupling and $H_2O$ dissociation[46,47]. The C–C coupling and $H_2O$ dissociation processes can be represented by

$$CO^* + CH_3^* \rightarrow CH_3CO^* \qquad (26)$$

$$H_2O^* \rightarrow H^* + OH^* \qquad (27)$$

For the theoretical energy band structure, the screened hybrid functional proposed by Heyd, Scuseria, and Ernzerhof (HSE)[48] was adopted to precisely calculate the DOS, from which band gap of $Cu_2Se$ and $Cu_2O$ can be obtained. The surface electrostatic potential is also computed to gain the work functions of $Cu_2Se$ and $Cu_2O$. Combining the DOS and work function, we can illustrate their band structure in theory.

**Finite-element method simulations.** The $CO_2$, $CH_3$, and CO concentrations were modeled by the COMSOL Multiphysics. The 2D reaction model was constructed with a platform represented for Cu foil and rectangular arrays represented for $Cu_2Se$ nanosheets covered with irregularly distributed circle represented for $Cu_2O$ particle (notably, the thickness of the sheets and the size of particles slightly larger than the actual size for a more intuitive comparison), a rectangular region filled with $CO_2$ was served as the calculated domain. The $CO_2$ was set as the feedstock diffused to the surface of the catalysts. All the meshes in the model were set to free tetrahedral meshing and the relative tolerance in the steady-state solver was set to 0.01.

In the simulation process, first, in the "Laminal Flow" module, the $CO_2$ was set to diffuse from the top boundary to the bottom boundary. Second, In the "Chemistry" module, three chemical species $CO_2$, $CH_3$, and CO was defined to symbolize the species involved in catalytic reactions. Five reactions were defined: three surface absorption-desorption equilibrium reactions for $CO_2$, $CH_3$, and CO, and two irreversible reactions for the $CO_2$ hydrogenating into $CH_3$ and $CO_2$ reducing into CO. Finally, the "Transport of Diluted Species (tds)" physics module was used to solve the mass transport of $CO_2$, $CH_3$, and CO. The species transport is driven by the diffusion, which follows Fick's law,

and the diffusion constants of $CO_2$, $CH_3$, and CO was taken to be to be $1.85 \times 10^{-9}$ m$^2$ s$^{-1}$, $1.85 \times 10^{-9}$ m$^2$ s$^{-1}$ and $1.00 \times 10^{-9}$ m$^2$ s$^{-1}$.

## Data availability
The data that support the plots within this paper and other findings of this study are available from the corresponding author upon reasonable request. Source data are provided with this paper.

## Code availability
The code that supports the findings of this study is available from the corresponding author upon request.

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

## Acknowledgements

This work was financially supported by European Research Council (ERC) under the European Union's Horizon 2020 research and innovation program (grant agreement No 819698 and GrapheneCore3: 881603), Deutsche Forschungsgemeinschaft (COORNETs, SPP 1928 and CRC 1415: 417590517), National Natural Science Foundation of China (22125503). The Supercomputing Center of Max Planck Computing & Data Facility (MPCDF) is acknowledged for computational support.

## Author contributions

X.F., Y.S., and X.D.L. conceived the idea and co-wrote the paper. X.D.L., L.L., G.C., X.H.L., X.C., Q.G., Z.Z., M.W. carried out the sample synthesis, characterization and $CO_2$ reduction measurement. X.D.L. and L.L. discussed the catalytic process. S.W. conducted the Finite-Element Method simulations. T.A. helped with the FIB-SEM characterizations. All the authors contributed to the overall scientific interpretation and edited the manuscript.

## Funding

## Competing interests

The authors declare no competing interests.
