## [Peer Review File · Nature Communications]

Photothermal CO₂ Conversion to Ethanol through Photothermal Heterojunction-Nanosheet ArraysREVIEWER COMMENTS

Reviewer #1 (Remarks to the Author):

In this manuscript, Li et al. reported a simple approach to synthesize a novel heterojunction-nanosheet array system for in situ thermally enhanced CO₂ photoconversion to ethanol. This approach exhibits huge potential for scale-up fabrication of the catalyst film and a record ethanol yield as high as 149.45 $\mu\text{mol g}^{-1} \text{h}^{-1}$ has been achieved by using the obtained array system. Furthermore, the authors carefully studied the catalytic reaction process, the role of each component, and the reaction mechanism by combining in situ characterization, theoretical calculations and controlled experiments under different conditions. All evidences are correctly characterized, and the manuscript is well structured and written. In this regard, this manuscript is of great importance and interest for the CO₂ conversion field, and can be accepted after minor revisions.

And, there are several tiny issues that should be concerned:

1. To reveal the role of the Cu₂O compound, the authors used the commercial Cu₂O powder as the comparison. But there seems to be a lack of corresponding characterization for the Cu₂O powder, like TEM. It should be better to fully characterize it.
2. Compared to the in situ grown Cu₂O on the surface Cu₂Se nanosheet, how about the performance of mechanically mixed catalytic system of Cu₂Se nanosheet and Cu₂O nanoparticle under the same reaction condition? What is the advantage of the in situ system?
3. Since the preparation of large-area Cu/Cu₂Se-Cu₂O heterojunction-nanosheet array system is possible, how about the potential of this system for scale-up CO₂ conversion?
4. We all know that H₂ evolution reaction is a strong competitive reaction during CO₂ conversion with H₂O as the proton source. How to avoid the production of H₂ in your case?

Reviewer #2 (Remarks to the Author):

In this paper, Li et al. reported an in situ thermally enhanced approach based on a Cu/Cu₂Se-Cu₂O heterojunction-nanosheet array catalyst system. The obtained Z-scheme Cu₂Se-Cu₂O heterostructure provides spatially separated redox sites for CO₂ reduction and water oxidation with boosted carrier transport efficiency. The resultant materials have been characterized by several different advanced techniques. This experimental work provides novel insights into design the unique catalyst for CO₂ photoconversion. I recommend publication of the paper after addressing the following issues.

1. How to control the amount of L-Cu₂Se nanosheet on Cu foil to get the best product?
2. In in situ grown H-Cu₂Se nanosheet on Cu foil, how to choose the ideal heating temperature and reaction time? Based on the existing conditions in the laboratory, or from the references?
3. The synthesis of Cu-CSCO HNA on Cu bulk and Cu-CSCO HNA on Cu foam, how to ensure that the same

product can grow on different substrates? How to control the thickness?

4. If the product of monoatomic layer is prepared on the substrate, what is the performance?
5. Compared with the reported results, what is the photocatalytic performance of the product?
6. In addition to in-situ infrared spectra, other in-situ data, such as in-situ Raman and in-situ TEM, can also be given.

Reviewer #3 (Remarks to the Author):

In this manuscript, the authors report an in situ thermally enhanced approach on Cu₂Se-Cu₂O heterostructure for CO₂ photoconversion, showing an impressive ethanol production with a rate of 149.45 μmol g⁻¹ h⁻¹, electron selectivity of 48.75% and AQE of 0.28%, under visible-near-infrared light irradiation without external heating. The microreactor formed by the gaps between Cu₂Se nanosheet arrays enhances the local concentration of intermediates (CH₃* and CO*), increasing the probability of C-C coupling which facilitate the generation of ethanol. By using photothermal materials to locally heat the catalytic region, the heat can effectively enhance the transfer of excited electrons, enabling them to surpass the reaction barrier. However, the paper lacks of solid evidence to support the conclusions and some necessary experiment and theoretical analyses must be supplemented:

1. According to the ideas presented in this article, the generation of hot electrons plays a crucial role in the reduction of CO₂ to ethanol. Do the authors have experimental results to demonstrate the role of hot electrons in this system? It would be beneficial for the authors to provide experimental evidence showing the generation of hot electrons in Cu₂Se under illumination and their function mechanism in CO₂ conversion.
2. The authors proposed that the localized high temperature generated by the photothermal material led to an increase in the concentrations of CH₃* and CO*, thereby promoting their coupling reactions. How did the authors prove that the localized high temperature increased the concentrations of CH₃* and CO*? Is there any experimental or theoretical evidence?
3. The theoretical calculations in Fig.4 demonstrated the role of Cu and CuSe forming a heterojunction in the C-C coupling, but they did not specifically address the increase in CH₃* and CO* concentrations due to the localized high temperature as mentioned by the authors. The authors should conduct additional in-depth analysis to investigate the specific mechanisms through which the localized high temperature influences the changes in CO₂ reduction pathways and its impact.
4. According to the yield results in Fig. 3c, CO is the main product with photo irradiation. The production of CH₄ increases, but CO decreases under thermal catalytic conditions. However, in this paper with photothermal, there's no CO production but a huge increase of CH₄ and CH₃CH₂OH. Why does the same catalyst exhibit significantly different CO₂ conversion pathways under photothermal condition compared with photo condition? The authors should further conduct a comprehensive reanalysis of the reaction mechanism for the high-selectivity conversion of CO₂ to ethanol, aiming to elucidate the factors influencing the formation of C₂ products, rather than solely proving the benefits of heterojunction formation based on theoretical calculations.
5. Based on the product results in Fig. 3c, the authors are supposed to reanalyze the mechanism of CO₂ photothermal conversion to ethanol and clarify the roles of Cu₂Se, Cu₂O and Cu foil, respectively.

Especially, Cu₂O has been widely discussed as an efficient active site for CO₂ reduction to C₂ product.

6. According to the proposed photogenerated charge carrier transfer pathway mechanism in Fig. 1c, the CO₂ reduction reaction is expected to occur on CuSe. What do the authors consider as the active site in this system? And what are the surface-adsorbed active species and intermediates during the CO₂ conversion to ethanol process?

7. Does the gap of the nanoarray favor the adsorption and capture of CO₂? The authors need to further investigate the catalyst's electron-donating ability towards CO₂ to support the study of the mechanism.

8. What is the valence state of Cu in this catalyst? Can the Cu⁺ stably exist during the CO₂ reduction process? Many studies have shown that the valence state and coordination environment of Cu in Cu-based catalysts have a significant impact on the CO₂ reduction performance. So, it's essential to provide the XPS or XAS of Cu.

9. The direction of the built-in electric field should be analyzed. Is there any experimental or theoretical results to support the function of Z-scheme heterostructure? Does Cu foil affect the built-in electric field and carrier migration? The result of photo-thermal catalysis for CO₂ reduction to ethanol is a cooperative effect of Cu₂Se, Cu₂O, and Cu foil. It would be biased to solely analyze the band structure between Cu₂Se and Cu₂O for its impact on CO₂ reduction.

10. According to Fig. 4b, the CO* peak at around 2100 cm⁻¹ grows with the increase of irradiation time. However, Fig. 3c shows that there's no CO generation. So, the authors should provide evidence to prove that CO* is difficult in desorption.

11. Since one species could have multiple absorption peaks and different species could have close IR absorptions, it is easier to cause misinterpretation if the authors did not comprehensively assess the possibility of different species and carefully search for their corresponding peaks. For instance, Peak 1 in Fig. 4b, which was assigned as CH₃* species in this manuscript, could also be -OH. And its wide wavelength range makes it more possible to be -OH from H₂O. So other stronger evidence should be provided to verify the formation of CH₃* and support the mechanism of CH₃*-CO* coupling.

12. Is there O₂ generation from water splitting? What's the reaction of holes? The generation rate of O₂ should be provided to clarify the reaction process between CO₂ and water.

13. What is the specific method for NMR testing? The configuration scheme for the testing liquid needs to be provided to facilitate readers in evaluating the appropriateness of ethanol content.

14. The authors emphasized the light response range of Cu-CSCO HNA from visible to NIR. Besides testing with 400 nm cutoff filter, did the authors study the performance of the catalyst under NIR light irradiation?

Responses to the Reviewers' comments and a summary of the changes made to the manuscript: NCOMMS-24-07495-T. We would like to thank all the reviewers for the insightful comments and suggestions, and for their time in helping us to improve this manuscript.

And we acknowledge the Reviewers' positive comments that "*this manuscript is of great importance and interest for the CO₂ conversion filed, and can be accepted after minor revisions*", "*This experimental work provides novel insights into design the unique catalyst for CO₂ photoconversion. I recommend publication of the paper after addressing the following issues*".

Point-to-point Response to Reviewer #1

Overall comments: *In this manuscript, Li et al. reported a simple approach to synthesize a novel heterojunction-nanosheet array system for in situ thermally enhanced CO₂ photoconversion to ethanol. This approach exhibits huge potential for scale-up fabrication of the catalyst film and a record ethanol yield as high as 149.45 $\mu\text{mol g}^{-1} \text{h}^{-1}$ has been achieved by using the obtained array system. Furthermore, the authors carefully studied the catalytic reaction process, the role of each component, and the reaction mechanism by combining in situ characterization, theoretical calculations and controlled experiments under different conditions. All evidences are correctly characterized, and the manuscript is well structured and written. In this regard, this manuscript is of great importance and interest for the CO₂ conversion filed, and can be accepted after minor revisions.*

Overall response: We appreciate the Reviewer's positive comments and constructive suggestions guiding the revision of our manuscript.

Comment 1: *To reveal the role of the Cu₂O compound, the authors used the commercial Cu₂O powder as the comparison. But there seems to be a lack of corresponding characterization for the Cu₂O powder, like TEM. It should be better to fully characterize it.*

Response: We appreciate the Reviewer's suggestions. We have added the corresponding characterizations of commercial Cu₂O powder into the revised version, like TEM, HRTEM, XRD and EDS mapping, as shown in Fig. N1 (Supplementary Figure 31):

Fig. N1 (Supplementary Figure 31) | Characterizations of commercial Cu_2O powder. (a) TEM image; (b) HRTEM image, in which the exposed facet can be inferred along [001] direction because 0.213 nm interplanar distances match well with the d_{020} and d_{200} spacings, and the corresponding dihedral angle of 90° agrees well with the calculated angle between the (020) and (200) planes; (c) XRD patterns; (d)-(f) annular dark-field TEM images and corresponding elemental mapping images.

Comment 2: Compared to the *in situ* grown Cu_2O on the surface Cu_2Se nanosheet, how about the performance of mechanically mixed catalytic system of Cu_2Se nanosheet and Cu_2O nanoparticle under the same reaction condition? What is the advantage of the *in situ* system?

Response: Thanks for the Reviewer's concerns. To explore the performance difference between the mechanically mixed system and *in situ* system, we also carried out the photocatalytic experiment of the mechanically mixed system of Cu_2Se nanosheet and commercial Cu_2O powder under the same reaction condition. The products were obtained as displayed in Fig. N2 (Supplementary Figure 40). From the GC spectrum, we can see that only a few CO ($1.24 \mu\text{mol g}^{-1} \text{h}^{-1}$) and CH_4 ($0.68 \mu\text{mol g}^{-1} \text{h}^{-1}$) gases were detected, and no liquid was produced according to the ^1H NMR spectrum.

Fig. N2 (Supplementary Figure 40) | The performance of mechanically mixed catalytic system of Cu₂Se nanosheet and commercial Cu₂O powder. The yield of (a) CO and (b) CH₄ with reaction time; (c) ¹H NMR spectrum of liquid products after 12 h reaction; (d) the generation rate of various products over this catalytic system.

The *in situ* growth of Cu₂O on the Cu₂Se nanosheet can ensure the close contact between these two compounds and forming the Z-scheme heterostructure, promoting the active electrons and holes participating into the CO₂ reduction and water oxidation, respectively. Meanwhile, the effective dynamic processes of the charge carrier can accelerate the separation and transport of the excited charge carriers, which can improve the performance of the Cu₂O-Cu₂Se heterostructure.

Comment 3: Since the preparation of large-area Cu/Cu₂Se-Cu₂O heterojunction-nanosheet array system is possible, how about the potential of this system for scale-up CO₂ conversion?

Response: As displayed in Fig.2a in our manuscript, we could easily fabricate the Cu/Cu₂Se-Cu₂O heterojunction-nanosheet array (Cu-CSCO HNA) system as large as 28 × 28 cm², but we don't have such a big sealed reactor to evaluate the photocatalytic performance of this large-area film. Actually, in our manuscript, we conducted photocatalytic experiments using the Cu-CSCO HNA system with different area sizes to explore the potential of this system for scale-up CO₂ conversion. As shown in Supplementary Fig. 28, we found that the yield of products using Cu-CSCO HNA for CO₂ photoreduction is linearly correlated to the catalyst area, suggesting that they are potentially scalable for large-area applications.

Comment 4: We all know that H_2 evolution reaction is a strong competitive reaction during CO_2 conversion with H_2O as the proton source. How to avoid the production of H_2 in your case?

Response: In our work, we used the gas-solid reactor to evaluate the performance of the Cu-CSCO HNA system as shown in Fig. N3 (Supplementary Figure 16). In this setup, the catalyst layer is separated from the water (proton source), which can prevent too much water from coming into contact with the catalyst, effectively avoiding competing hydrogen evolution reactions (HER).

Fig. N3 (Supplementary Figure 16) | The setup of photocatalytic reactor in our work, in which the Cu-CSCO HNA film is supported by a hollow quartz column to avoid direct contact between catalyst and water. (a) Schematic diagram; (b) digital image.

Point-to-point Response to Reviewer #2

Overall comments: In this paper, Li et al. reported an *in situ* thermally enhanced approach based on a Cu/Cu₂Se-Cu₂O heterojunction-nanosheet array catalyst system. The obtained Z-scheme Cu₂Se-Cu₂O heterostructure provides spatially separated redox sites for CO₂ reduction and water oxidation with boosted carrier transport efficiency. The resultant materials have been characterized by several different advanced techniques. This experimental work provides novel insights into design the unique catalyst for CO₂ photoconversion. I recommend publication of the paper after addressing the following issues.

Overall response: Thanks a lot for the Reviewer's positive comments and kind suggestions.

Comment 1: How to control the amount of L-Cu₂Se nanosheet on Cu foil to get the best product?

Response: Thanks for the Reviewer's concerns. To get the best photothermal effect, we hope that the Cu₂Se nanosheet completely and uniformly covers the surface of Cu foil. To achieve this purpose, we controlled the amount of L-Cu₂Se nanosheet on Cu foil by regulating the reaction time and Se precursor solution concentration. As displayed in Fig. N4 (Supplementary Figure 3), too short a reaction time could not completely cover the Cu foil surface with Cu₂Se nanosheet.

Fig. N4 (Supplementary Figure 3) | The SEM images of L-Cu₂Se nanosheets on Cu foil with different reaction time. (a)-(b) 0.5 h; (c)-(d) 1 h; (e)-(f) 1.5 h.

Besides, we actually also explored the effect of Se precursor solution concentration on the growth of Cu₂Se nanosheet. As shown in Fig. N5 (Supplementary Figure 4), we found that the lower concentration of Se precursor solution compared to the concentration we used in the manuscript can cause the presence of impurities while the higher concentration has basically no effect on the growth of Cu₂Se nanosheet. Therefore, we chose the reaction condition described in our manuscript to prepare the L-Cu₂Se nanosheet on Cu foil.

Fig. N5 (Supplementary Figure 4) | The SEM images of L-Cu₂Se nanosheets on Cu foil with different concentration of Se precursor solution. (a)-(b) 0.5 times compared to the concentration used in the manuscript; (c)-(d) 2 times compared to the concentration used in the manuscript.

Comment 2: *In situ* grown H-Cu₂Se nanosheet on Cu foil, how to choose the ideal heating

temperature and reaction time? Based on the existing conditions in the laboratory, or from the references?

Response: To get the ideal heating temperature and reaction time for *in situ* growth of H-Cu₂Se nanosheet on Cu foil, we actually tried the calcination process at different conditions. For example, we calcinated the L-Cu₂Se nanosheet at 100 and 150 °C for 45 mins, respectively. After that, there was a mixed crystal phase with L-Cu₂Se and H-Cu₂Se as shown in Fig. N6a-b, which means that the L-Cu₂Se can't be totally transformed into H-Cu₂Se completely at the low temperature. Meanwhile, the short reaction time also led to the appearance of mixed phases as displayed in Supplementary Figure 7 and Fig. N6c. Of course, the higher temperature and longer reaction time also work for the formation of H-Cu₂Se nanosheet on Cu foil, to minimize the energy consumption, we chose those conditions in our manuscript (200 °C for 45 mins with a heating rate of 10 °C min⁻¹).

Fig. N6 (Supplementary Figure 6) | XRD patterns of Cu-Cu₂Se under different conditions. (a) Cu-Cu₂Se after 100 °C for 45 mins. (b) Cu-Cu₂Se after 150 °C for 45 mins. (c) Cu-Cu₂Se after 200 °C for 20 mins.

Comment 3: The synthesis of Cu-CSCO HNA on Cu bulk and Cu-CSCO HNA on Cu foam, how to ensure that the same product can grow on different substrates? How to control the thickness?

Response: Thanks for the Reviewer's concerns. To ensure that the same Cu-CSCO HNA grow on different substrates like Cu bulk and Cu foam, we used the same reaction solution and reaction time with that in synthesis of Cu-CSCO HNA on Cu foil, which would make sure that the Cu₂Se nanosheets completely cover and uniformly grow on the substrates. We also carried out the control experiments for preparing the Cu-CSCO HNA on Cu bulk and Cu foam under different reaction times (Fig. N7), from which the same phenomenon was observed with that for synthesis of Cu-CSCO HNA on Cu foil. Only when the reaction time is long enough, we could get the totally covered Cu-CSCO HNA. Honestly, we only tried to control the nanosheet morphology and the array structure; there might be some difference in the size of the nanosheet on different substrates, which could not be avoided during the synthesis.

Fig. N7 | The SEM images of Cu_2Se nanosheets on Cu foam and bulk with different reaction time. (a) 0.5 h on Cu bulk. (b) 1 h on Cu bulk. (c) 1.5 h on Cu bulk. (d) 0.5 h on Cu foam. (e) 1 h on Cu foam. (f) 1.5 h on Cu foam.

To control the thickness, we tried to investigate the thickness of the Cu-CSCO HNA on Cu foil under different growth conditions. As shown in Fig. N8, we found that thickness of Cu-CSCO HNA on Cu foil is almost the same as that of the sample synthesized in our manuscript when using a higher concentration of Se precursor solution or a longer reaction time. This is the reason why we chose the synthesis method described in the method section. As long as the concentration of the Se precursor solution is high enough, the growth solution is no longer in contact with the Cu substrate when the Cu_2Se nanosheets completely cover the Cu surface, and the thickness basically does not increase with the reaction time.

Fig. N8 | The SEM section images of Cu-CSCO HNA on Cu foil with different reaction conditions. (a) The used Cu-CSCO HNA on Cu foil in our manuscript. (b) The Cu-CSCO HNA on Cu foil using 2 times concentration of Se precursor solution. (c) The Cu-CSCO HNA on Cu foil after 4h growing.

Comment 4: *If the product of monoatomic layer is prepared on the substrate, what is the performance?*

Response: Thanks for the Reviewer's concerns. In our case, it is impossible to achieve the monoatomic Cu_2Se layer on the substrates. As shown in Fig. N4, the Cu_2Se nanosheets don't grow on the Cu foil through layer by layer. That is to say, we can only get the vertically separated Cu_2Se nanosheets instead of the monoatomic Cu_2Se layer on the substrate under a shorter reaction time.

Comment 5: Compared with the reported results, what is the photocatalytic performance of the product?

Response: Thanks for the Reviewer's concerns. We compared our photocatalytic performance with the reported results in the manuscript. As shown in Fig. 3d and Supplementary Figure 22, the achieved rate of ethanol generation is nearly 3 times higher than the state-of-the-art performance in the photo and photothermal catalysis, while the ethylene yield is also superior to the most of the reported photocatalysts.

Comment 6: In addition to in-situ infrared spectra, other in-situ data, such as in-situ Raman and in-situ TEM, can also be given.

Response: According to Reviewer's suggestion, we have performed in-situ Raman to detect the reaction processes. The corresponding results are shown in Fig. N9 (Supplementary Figure 53). The peaks at around 2040 and 2100 cm^{-1} are attributed to $\nu(\text{CO})$ while the peaks at 2805 and 2924 cm^{-1} are assigned to the characteristic bands of $\nu(\text{CH}_3)$ (*J. Chem. Phys.* 2019, 150, 041718; *J. Phys. Chem. A* 2013, 117, 4377-4384; *J. Quant. Spectrosc. Ra.* 2022, 277, 107978), both of which are crucial initial intermediates for ethanol generation.

Fig. N9 (Supplementary Figure 53) | *In situ* Raman spectra characterization for co-adsorption of a mixture of CO_2 and H_2O vapor under light irradiation over Cu-CSCO HNA system.

There is no suitable in-situ TEM setup for photocatalysis so far. We are sorry we could not provide in-situ TEM data here.

Point-to-point Response to Reviewer #3

Overall comments: In this manuscript, the authors report an in situ thermally enhanced approach on $\text{Cu}_2\text{Se-Cu}_2\text{O}$ heterostructure for CO_2 photoconversion, showing an impressive ethanol production with a rate of $149.45 \mu\text{mol g}^{-1} \text{h}^{-1}$, electron selectivity of 48.75% and AQE of 0.28%, under visible-near-infrared light irradiation without external heating. The microreactor formed by the gaps between Cu_2Se nanosheet arrays enhances the local concentration of intermediates (CH_3^* and CO^*), increasing the probability of C-C coupling

which facilitate the generation of ethanol. By using photothermal materials to locally heat the catalytic region, the heat can effectively enhance the transfer of excited electrons, enabling them to surpass the reaction barrier. However, the paper lacks of solid evidence to support the conclusions and some necessary experiment and theoretical analyses must be supplemented.

Overall response: We appreciate the Reviewer's comments and kind suggestions. To support the conclusions, we have provided more theoretical and experimental evidence in the revised version, such as controlled experiments, adsorption isotherms for different gases, TPD, XPS, *in situ* Raman, finite element analysis, and DFT calculations.

Comment 1: *According to the ideas presented in this article, the generation of hot electrons plays a crucial role in the reduction of CO₂ to ethanol. Do the authors have experimental results to demonstrate the role of hot electrons in this system? It would be beneficial for the authors to provide experimental evidence showing the generation of hot electrons in Cu₂Se under illumination and their function mechanism in CO₂ conversion.*

Response: Thanks for the Reviewer's valuable comment. For the issue of hot electrons, actually we did not mention them playing a crucial role in the reduction of CO₂ to ethanol. Normally, the hot electrons are more likely produced in the metallic materials, like Au, Ag, by surface plasmon resonance effect (*Nat. Commun.* 2020, 11, 1615; *Science* 2015, 349,632-635; *Nat. Photon.* 2015, 9, 471-476), which doesn't quite fit into our semiconductor system. In our work, the photothermal effect indeed plays a crucial role. In this process, light illumination excites the valence band electrons of Cu₂Se nanosheets to transition to higher energy levels in the conduction band, generating high-energy electrons. These high-energy electrons transfer energy to the material lattice through nonradiative relaxation via Auger recombination, leading to a local temperature rise in the lattice (*Phys. Rev.* 1961, 122, 419-424; *Energy Environ. Sci.* 2019, 12, 1122-1142). Typically, the energetic carriers thermalize back to the band edge through lattice vibrations, further participating into the catalytic reaction (*Chem. Rev.* 2023, 123, 6891-6952). The remarkable photothermal effect can be confirmed from the photothermal curves and images in Fig. 3b and Supplementary Figure 14, in which the Cu-CSCO HNA system is able to elevate its temperature up to approximately 200 °C in just 2 minutes. And, if we exclude the influence of heat, as shown in the Fig. 3C and Supplementary Figure 16 c-d, the performance of ethanol generation is much lower than that under photothermal condition.

Comment 2: *The authors proposed that the localized high temperature generated by the photothermal material led to an increase in the concentrations of CH₃* and CO*, thereby promoting their coupling reactions. How did the authors prove that the localized high temperature increased the concentrations of CH₃* and CO*? Is there any experimental or theoretical evidence?*

Response: Thanks for the Reviewer's comment. We are sorry for the possible confusion here. Actually, we did not claim the localized high temperature generated by the photothermal material could lead to an increase in the concentrations of CH₃* and CO*. In fact, we claimed in the manuscript that “the microreactors induced by Cu₂Se nanosheet array gaps improve the local concentration of intermediates (CH₃* and CO*), thereby increasing the probability of further C-C coupling.” We confirmed the array structure increasing the intermediate concentration by *in situ* FTIR spectroscopy (Fig. 4b and Supplementary Figure 52), in which

the Cu-CSCO HNA system (with array structure) exhibited higher peak intensity of CO^* , CH_3^* and $\text{CH}_3\text{CH}_2\text{O}^*$ species than that in the CSCO heterojunction powder sprayed on Cu foil (without array structure).

To make it more solid, we further calculated the concentration distribution of CH_3^* and CO^* intermediates using finite element analysis. As displayed in Fig. N10 (Supplementary Figure 48), the microreactors induced by Cu_2Se nanosheet array gaps could not only enhance the CO_2 adsorption on the catalyst surface, but also increase the local concentration of CH_3^* and CO^* intermediates, which would improve the C-C coupling and promote the generation of ethanol.

Fig. N10 (Supplementary Figure 48) | Finite-element method simulations. (a) The size of the theoretical models. (b) Concentration distribution of CO_2 . (c) Concentration distribution of CH_3^* specie. (d) Concentration distribution of CO^* specie.

Comment 3: *The theoretical calculations in Fig.4 demonstrated the role of Cu and Cu_2Se forming a heterojunction in the C-C coupling, but they did not specifically address the increase in CH_3^* and CO^* concentrations due to the localized high temperature as mentioned by the authors. The authors should conduct additional in-depth analysis to investigate the specific mechanisms through which the localized high temperature influences the changes in CO_2 reduction pathways and its impact.*

Response: Thanks for the Reviewer's concerns. Like the last comment, we did not claim the increase of CH_3^* and CO^* concentrations is due to the localized high temperature. Instead, we claimed that the microreactors induced by Cu_2Se nanosheet array gaps actually improve the local concentration of intermediates (CH_3^* and CO^*).

Comment 4: According to the yield results in Fig. 3c, CO is the main product with photo irradiation. The production of CH₄ increases, but CO decreases under thermal catalytic conditions. However, in this paper with photothermal, there's no CO production but a huge increase of CH₄ and CH₃CH₂OH. Why does the same catalyst exhibit significantly different CO₂ conversion pathways under photothermal condition compared with photo condition? The authors should further conduct a comprehensive reanalysis of the reaction mechanism for the high-selectivity conversion of CO₂ to ethanol, aiming to elucidate the factors influencing the formation of C₂ products, rather than solely proving the benefits of heterojunction formation based on theoretical calculations.

Response: We appreciate the valuable comments from the reviewer. Actually we have provided a detailed analysis in the manuscript regarding the selective variation of products under conditions of photothermal, photonic, and purely thermal effects (Page7, line 4-5 and 16-18; Page9, line 28-34 and 40-44). Under photo condition excluding the heat, the photoexcited electrons can transfer to the surface of Cu₂Se nanosheet to participate into the CO₂ reduction. But without the local heat, the energy barrier of the protonation processes would hinder the deep reduction of CO₂ molecules, resulting in only the generation of CO product. When adding the heat by the photothermal effect, the heterostructure could accelerate the photoexcited carrier separation and transport, which could ensure abundant electrons participate into CO₂ reduction. Meanwhile, the high temperature can enhance the probability of reactants overcoming the activation energy by skewing the Boltzmann distribution towards higher energies (*Chem. Rev.* 2023, 123, 6891-6952; *Chem. Sci.* 2016, 7, 6887-6893), which is beneficial for the formation of C-H bonds during the CO₂ conversion processes (*Nat. Catal.* 2023, 6, 519-530). Besides, the gaps induced by the nanosheet array structure can limit release of the reaction intermediates (CO* and CH₃*) and increase their concentration for C-C coupling. Therefore, we could achieve a huge increase of CH₄ and CH₃CH₂OH using the system under the photothermal condition.

To confirm the above reaction mechanism, we indeed carried out a lot of controlled experiments and theoretical calculations. For example, we compared the performance of Cu-CSCO HNA system with and without the heat (Supplementary Figure 16), the array structure (Supplementary Figure 41-43) and the heterostructure (Supplementary Figure 40). We also calculated the adsorption energy of reaction intermediates (Supplementary Figure 47, 51) and the C-C coupling energy barrier (Supplementary Figure 54-61) on different surface and under different CO* concentrations.

Besides, to further confirm the reaction mechanism, we've added additional controlled experiments. We used the mechanical friction method to flat the vertical nanosheets to eliminate gaps in the Cu-CSCO HNA system as shown in Fig. N11. With this Cu-CSCO HNA system without gap for photothermal CO₂ reduction, there is only trace C₂H₄ and ethanol product detected, confirming that the gap induced by the nanosheet array is indeed contributed to the generation of multi-carbon products (Fig. N12). And we also added the finite-element method simulations (Fig. N10) to confirm the localization effect of the array structure for increasing the reaction intermediates concentration as described in Comment 2.

To make the mechanism easier to understand, we have changed our description in the revised version and formulated it more clearly.

Fig. N11 (Supplementary Figure 41) | Cu-CSCO HNA treated by mechanical friction. (a) The scheme of the mechanical friction process. (b) Digital images of Cu-CSCO HNA with (blue box) and without gap (red box). (c) Digital images of Cu-CSCO HNA without gap for catalysis experiment. (d) SEM image.

Fig. N12 (Supplementary Figure 42) | The performance of Cu-CSCO HNA without gap for photothermal CO₂ reduction. (a) GC of the gas products, in which the CO, CH₄ and C₂H₄ were detected as the reduction products while the O₂ was the oxidation product. (b) ¹H NMR spectra for liquid product, in which DMSO was used as the reference. (c) The yield of each product (CO: 5.44 μmol g⁻¹ h⁻¹; CH₄: 86.32 μmol g⁻¹ h⁻¹; C₂H₄: 0.81 μmol g⁻¹ h⁻¹; ethanol: 12.17 μmol g⁻¹ h⁻¹).

Comment 5: Based on the product results in Fig. 3c, the authors are supposed to reanalyze the mechanism of CO₂ photothermal conversion to ethanol and clarify the roles of Cu₂Se, Cu₂O and Cu foil, respectively. Especially, Cu₂O has been widely discussed as an efficient active site for CO₂ reduction to C₂ product.

Response: Thanks for the Reviewer's concerns. We've already synthesized the separated Cu₂Se nanosheet, Cu₂O nanoparticle and the Cu₂Se-Cu₂O heterostructure without the Cu substrate. To further clarify the role of Cu₂Se, Cu₂O and Cu foil respectively, we also added the comparative experiments using pure Cu₂O nanoparticle and pure Cu foil as the catalyst for CO₂ reduction. As displayed in Fig. N13, both the Cu₂O and pure Cu foil showed a trace amount of CH₄ generation (10.77 μmol g⁻¹ h⁻¹ for Cu₂O nanoparticle and 0.68 μmol g⁻¹ h⁻¹ for pure Cu foil) and no liquid product was detected.

Fig. N13 | The photocatalytic performance of (a) Cu₂O nanoparticle and (b) pure Cu foil for CO₂ reduction including the heat.

For the performance of pure Cu₂Se nanosheet, we found that it would be partially oxidized to Cu₂O by O₂ during the photothermal catalysis, which was confirmed by the XRD pattern after catalysis (Fig. N14). Therefore, it's really hard to analyze the role of pure Cu₂Se nanosheet individually.

Fig. N14 | (a) Setup for photothermal CO₂ reduction over pure Cu₂Se nanosheet. (b) XRD pattern of Cu₂Se nanosheet after photothermal catalysis.

Based on our existing experimental results, we can conclude that Cu₂Se acts as a photothermal material, generating localized high temperatures. The heterojunction formed between Cu₂O and Cu₂Se accelerates carrier separation and transport on one hand, and promotes the redox reaction on the other hand. Additionally, Cu serves as the substrate, maintaining localized high temperatures, while interface charges can enhance intermediate adsorption, accelerate C-C coupling, and promote the generation of multiple carbons. We have modified the corresponding description to make it clearer in the revised version.

Regarding what the reviewer mentioned that “Cu₂O has been widely discussed as an efficient active site for CO₂ reduction to C₂ product”, we have consulted numerous literatures. We found that Cu₂O can only be used as the active compound for CO₂ reduction to C₂ products under electrocatalysis. For photocatalytic CO₂ reduction, there is no report for Cu₂O generating C₂ products (*J. Mater. Chem. A*, 2021, 9, 5915-5951; *Nat Energy* 2019, 4, 957-968). And we also tried the CO₂ photoreduction experiments using the pure Cu₂O as the catalyst. As shown in Fig. N13a, we only detected trace CH₄ produced during the photocatalysis, well-consist with the previous reports (*Angew. Chem. Int. Ed.* 2021, 60, 8455).

Comment 6: According to the proposed photogenerated charge carrier transfer pathway mechanism in Fig. 1c, the CO₂ reduction reaction is expected to occur on Cu₂Se. What do the authors consider as the active site in this system? And what are the surface-adsorbed active species and intermediates during the CO₂ conversion to ethanol process?

Response: Thanks for the Reviewer’s concerns. For Cu-based materials for CO₂ photoreduction, in the majority of literatures, Cu is commonly regarded as the active site due to its vacant orbitals for intermediate adsorption and occupied orbitals for electron donation (*Angew. Chem. Int. Ed.* 2022, e202216613; *Angew. Chem. Int. Ed.* 2023, e202216717). We also calculated the adsorption energy of CO* intermediate at Cu and Se atoms to check the active site. As shown in Supplementary Figure 47d-e, we found that the surface Cu atom possessed stronger adsorption energy than Se atom for CO* species. Thus we considered Cu as the active site in our system.

To detect the surface-adsorbed active species and intermediates during the CO₂ conversion to

ethanol process, we carried out *in situ* Fourier-transform infrared (FTIR) spectroscopy. As displayed in Fig. 4b, all the COOH*, CO*, CHO*, CH₃O*, CH₃*, CH₃CH₂O* could be the reaction intermediates for CO₂ reduction. According to the *in situ* FTIR spectroscopy, the possible reaction pathway was speculated as described from equation (1) to (9) in our manuscript, which is quite in agreement with that in the previous reports for ethanol generation (*J. Am. Chem. Soc.* 2023, 145, 15343-15352; *J. Am. Chem. Soc.* 2022, 144, 20495-20506).

Comment 7: Does the gap of the nanoarray favor the adsorption and capture of CO₂? The authors need to further investigate the catalyst's electron-donating ability towards CO₂ to support the study of the mechanism.

Response: We appreciate the valuable comment from the reviewer. In fact, the gap of the nanoarray indeed favors the adsorption and capture of CO₂. To confirm that, we first conducted the finite element analysis. As displayed in Fig. N10 (Supplementary Figure 48), the gaps induced by Cu₂Se nanosheet array can not only enhance the CO₂ adsorption on the catalyst surface, but also increase the local concentration of CH₃* and CO* intermediates, which could improve the C-C coupling and promote the generation of ethanol. And we also carried out the CO₂ adsorption isotherms for Cu-CSCO HNA with and without gap treated by mechanical friction (Fig. N15). Since the whole Cu foil can't be used for the CO₂ adsorption experiment directly, we then cut it into the little species to conduct the test, which does not infect the gap structure. As displayed in Fig. N15 (Supplementary Figure 49), Cu-CSCO HNA with gap exhibits a better capability for CO₂ adsorption than that without gap, further confirming the gap of the nanoarray favors the adsorption and capture of CO₂

Fig. N15 (Supplementary Figure 49) | CO₂ adsorption isotherms for Cu-CSCO HNA with (red) and without gap (black).

Comment 8: What is the valence state of Cu in this catalyst? Can the Cu⁺ stably exist during the CO₂ reduction process? Many studies have shown that the valence state and coordination environment of Cu in Cu-based catalysts have a significant impact on the CO₂ reduction performance. So, it's essential to provide the XPS or XAS of Cu.

Response: We thank the Reviewer's thoughtful comments. We have already provided the XPS spectra of Cu-CSCO HNA (Supplementary Figure 10) in our manuscript. But since the peaks of Cu^0 and Cu^+ are located at the same position, we can hardly determine the valence state of Cu according to the provided XPS. So we further conducted the XPS tests to get the Cu LMM Auger electron spectroscopy to identify its valence state. As displayed in Fig. N16 (Supplementary Figure 25), we found that only the peaks of Cu^+ at around 916.7 eV were detected on the surface of Cu-CSCO HNA before and after photothermal catalysis, which demonstrates that the valence state of Cu in this catalyst is +1, and the Cu^+ can stably exist during the CO_2 reduction process.

Fig. N16 (Supplementary Figure 25) | Cu LMM Auger electron spectroscopy of Cu-CSCO HNA before (black) and after (red) photothermal catalysis, in which only the peaks of Cu^+ were detected at around 916.7 eV.

Comment 9: *The direction of the built-in electric field should be analyzed. Is there any experimental or theoretical results to support the function of Z-scheme heterostructure? Does Cu foil affect the built-in electric field and carrier migration? The result of photo-thermal catalysis for CO_2 reduction to ethanol is a cooperative effect of Cu_2Se , Cu_2O , and Cu foil. It would be biased to solely analyze the band structure between Cu_2Se and Cu_2O for its impact on CO_2 reduction.*

Response: Thanks for the Reviewer's comment. To analyze the direction of the built-in electric field, we have calculated the band structure of Cu_2Se and Cu_2O in theory and also confirmed their band edge positions by combining the UPS and UV-vis-NIR diffuse reflectance spectra. According to the theoretical results in Fig. 1, the band edge positions of Cu_2Se and Cu_2O meet the requirement for forming the Z-scheme heterostructure and the direction of the built-in electric field is from Cu_2Se (positive charge) to Cu_2O (negative charge), which is well-consistent with the experimental results in Supplementary Figure 34. In this Z-scheme heterostructure, the photoexcited electrons in Cu_2O can be transported to the Cu_2Se to realize CO_2 reduction reaction while the remained holes in Cu_2O can participate into the water oxidation simultaneously. To confirm that, we have further conducted the XPS spectra of Cu_2Se , Cu_2O and the heterostructure to explore the charge transfer. As shown in Supplementary Figure 36, compared with the separated Cu_2Se nanosheet and Cu_2O nanoparticle, the XPS peak of Se in heterostructure shifted to lower binding energy while the O peak in heterostructure showed

an obvious offset to higher binding energy, which strongly demonstrated the electron transfer from the Cu_2O to Cu_2Se after forming the heterostructure of these two compounds.

To explore the impact of Cu foil on the charge carrier migration, we conducted additional XPS tests for Cu- Cu_2Se system without Cu_2O and compared it with the XPS spectra of pure Cu_2Se and CSCO powder (Supplementary Figure 36) as well as Cu-CSCO HNA (Supplementary Figure 10). As displayed in Fig. N17, comparing these peaks of Se, we found that the binding energy of the Se peaks showed an obvious trend, $\text{Cu-Cu}_2\text{Se} > \text{Cu-CSCO HNA} > \text{Cu}_2\text{Se} > \text{CSCO powder}$, which confirmed that electrons could transfer from Cu_2O to Cu_2Se (due to Se peak in $\text{Cu}_2\text{Se} > \text{CSCO powder}$) and also moved from Cu_2Se to Cu foil (due to Se peak in $\text{Cu-Cu}_2\text{Se} > \text{Cu}_2\text{Se}$ and $\text{Cu-CSCO HNA} > \text{CSCO powder}$). Meanwhile, the binding energy of the O peaks exhibited a trend of $\text{Cu-CSCO HNA} > \text{CSCO powder} > \text{Cu}_2\text{O}$, which further demonstrated the electron transfer from Cu_2O to Cu_2Se (due to O peak in $\text{CSCO powder} > \text{Cu}_2\text{O}$) and also from Cu_2Se to Cu foil (due to O peak in $\text{Cu-CSCO HNA} > \text{CSCO powder}$). Since the O peak in Cu-CSCO HNA was shifted more than that in CSCO powder compared with the pure Cu_2O , we can conclude that the Cu foil doesn't change the direction of the built-in electric field but further promotes the carrier migration in Cu-CSCO HNA system.

Fig. N17 | XPS spectra of Cu_2O , Cu_2Se , Cu- Cu_2Se , CSCO powder and Cu-CSCO HNA. (a) High-resolution Se 3d spectra. (b) High-resolution O 1s spectra.

For the cooperative effect of Cu_2Se , Cu_2O , and Cu foil, we did not “solely analyze the band structure between Cu_2Se and Cu_2O for its impact on CO_2 reduction”. Just as replied in comment 4, we actually carried out a lot of controlled experiments and theoretical calculations. For example, we compared the performance of Cu-CSCO HNA system with and without the heat (Supplementary Figure 16), the array structure (Supplementary Figure 41-43) and the heterostructure (Supplementary Figure 40). We also calculated the adsorption energy of reaction intermediates (Supplementary Figure 47, 51) and the C-C coupling energy barrier (Supplementary Figure 54-61) on different surface and under different CO^* concentrations. Besides, we further added additional controlled experiments. We used the mechanical friction method to flat the vertical nanosheets to eliminate gaps in the Cu-CSCO HNA system to explore the impact of gaps on the generation of multi-carbon products (Fig. N11). And we also added

the finite-element method simulations (Fig. N10) to confirm the localization effect of the array structure for increasing the reaction intermediates concentration as described in Comment 2.

Comment 10: According to Fig. 4b, the CO* peak at around 2100 cm^{-1} grows with the increase of irradiation time. However, Fig. 3c shows that there's no CO generation. So, the authors should provide evidence to prove that CO* is difficult in desorption.

Response: Thanks for the Reviewer's comment. To prove that CO* is difficult in desorption in the Cu-CSCO HNA system, we have calculated the CO* adsorption on pure Cu_2Se surface and at the interface of Cu- Cu_2Se . As shown in Supplementary Figure 47, a stronger CO adsorption energy of -0.89 eV is obtained at the Cu- Cu_2Se interface than that on the pure Cu_2Se surface (-0.68 eV), suggesting that the CO* desorption in the Cu-CSCO HNA system is more difficult. Besides, we also calculated the concentration distribution of CO* intermediates using finite element analysis as described in Comment 2. As displayed in Fig. N10, the microreactors induced by Cu_2Se nanosheet array gaps can limit CO* desorption and increase its local concentration. Furthermore, we added the CO temperature-programmed desorption (TPD) measurement of Cu-CSCO HNA with and without gap. As shown in Fig. N18 (Supplementary Figure 50), the temperature of CO desorption for the Cu-CSCO HNA without gap (treated by mechanical friction method as shown in Fig. N11) is 136 °C, which is much smaller than that for the Cu-CSCO HNA with gap (213 °C), confirming that the CO desorption is harder in the Cu-CSCO HNA system with gap.

Fig. N18 (Supplementary Figure 50) | CO-TPD measurement. The temperature of CO desorption for the Cu-CSCO HNA without gap (treated by mechanical friction method as shown in Fig. N11) is 136 °C, which is much smaller than that for the Cu-CSCO HNA with gap (213 °C), confirming the CO desorption is harder in the Cu-CSCO HNA system with gap.

Comment 11: Since one species could have multiple absorption peaks and different species could have close IR absorptions, it is easier to cause misinterpretation if the authors did not comprehensively assess the possibility of different species and carefully search for their

corresponding peaks. For instance, Peak 1 in Fig. 4b, which was assigned as CH_3^* species in this manuscript, could also be $-\text{OH}$. And its wide wavelength range makes it more possible to be $-\text{OH}$ from H_2O . So other stronger evidence should be provided to verify the formation of CH_3^* and support the mechanism of $\text{CH}_3^*-\text{CO}^*$ coupling.

Response: Thanks for the Reviewer's concerns. We do agree that "one species could have multiple absorption peaks and different species could have close IR absorptions". Regarding the reviewer's concern that the peak of CH_3^* could be $-\text{OH}$ from H_2O , we don't think it is possible in our case. Normally, the IR peaks for $-\text{OH}$ from H_2O should be more than 3400 cm^{-1} (*Appl. Surf. Sci.* 2016, 389, 721-734; *J. Hazard. Mater.* 2019, 364, 100-107) while the peaks of CH_3^* had been confirmed at around $2920\text{--}2915\text{ cm}^{-1}$ according to previous reports (*Arab. J. Chem.* 2020, 13, 851-862; *Catal. Today* 2012, 182, 3-11), well-consistent with the result in our work. To further verify the formation of CH_3^* instead of $-\text{OH}$, we also conducted *in situ* FTIR spectra using the Cu-CSCO HNA system with Ar and water. As shown in Fig. N19, the peaks from $3500\text{--}3800\text{ cm}^{-1}$ were induced by the $-\text{OH}$ species from H_2O while no any signal of CH_3^* at 2920 cm^{-1} was detected. Since it is very difficult to completely remove the CO_2 by Ar flow, the peaks at around 2360 cm^{-1} were identified from the trace CO_2 in the system.

Fig. N19 | *In situ* FTIR spectroscopy characterization for co-adsorption of a mixture of Ar and H_2O vapor under light irradiation over Cu-CSCO HNA. The peaks at around 2360 cm^{-1} come from the trace CO_2 in the system. And the peaks from $3500\text{--}3800\text{ cm}^{-1}$ are induced by the $-\text{OH}$ species from H_2O while no signal of CH_3^* at 2920 cm^{-1} was detected.

Apart from *in situ* FTIR spectra, we further added *in situ* Raman spectra to explore the reaction mechanism. As shown in Fig. N9, the peaks at around 2040 and 2100 cm^{-1} are attributed to $\nu(\text{CO})$ while the peaks at 2805 and 2924 cm^{-1} are assigned to the characteristic bands of $\nu(\text{CH}_3)$ (*J. Chem. Phys.* 2019, 150, 041718; *J. Phys. Chem. A* 2013, 117, 4377-4384; *J. Quant. Spectrosc. Ra.* 2022, 277, 107978), both of which are crucial initial intermediates for ethanol generation.

Comment 12: Is there O_2 generation from water splitting? What's the reaction of holes? The

generation rate of O_2 should be provided to clarify the reaction process between CO_2 and water.

Response: Thanks for the Reviewer's concerns. Yes, there is O_2 generation from water splitting in our case and we have already described this in our manuscript, "... while O_2 is detected as the oxidation product". The separated holes indeed participated into the water oxidation reaction to O_2 . The produced O_2 was confirmed by the gas chromatography (GC) as shown in Supplementary Figure 18. To make it more precise, we further added the generation rate of O_2 during photothermal catalysis in the revised version. As shown in Fig. N20 (Supplementary Figure 19), the obtained yield for O_2 generation is around $632.96 \mu\text{mol g}^{-1} \text{h}^{-1}$. It is worth noting that the consumed holes calculated from oxygen yield are smaller than the electrons consumed by CO_2 reduction. That's because a lot of the oxygen produced will be dissolved in the solution and difficult to detect.

Fig. N20 (Supplementary Figure 19) | The yield of O_2 during photothermal catalysis Cu-CSCO HNA with irradiation time.

Comment 13: What is the specific method for NMR testing? The configuration scheme for the testing liquid needs to be provided to facilitate readers in evaluating the appropriateness of ethanol content.

Response: Thanks for the Reviewer's concerns. For detecting the liquid products by NMR, the specific operation methods are as follows: We first took $400 \mu\text{L}$ solution from the bottom of the reactor after the catalysis and mixed it with $100 \mu\text{L}$ deuterated water (D_2O). Then $20 \mu\text{L}$ dimethyl sulfoxide (DMSO) (diluted 10,000 times with the concentration of 1.413 mM) was added as internal standards for the ^1H NMR analysis. The internal standards, DMSO, was chosen because they did not interfere with peaks arising from CO_2 reduction products and because of their non-volatility which allowed for use and storage of the same internal standards solution for all of the product measurements without appreciable change in concentration. The area of product peaks was compared to the area of DMSO (at a chemical shift of 2.60 ppm). In our case, the triple peak of ethanol at a chemical shift of 1.06 ppm was used to calculate the generation rate (N) as the following equation:

$$N = \frac{S \times V3 \times n \times 6 \times V2}{3 \times V1 \times t \times m}$$

where S is the area of triple peak of ethanol compared to DMSO (identified to 1 as reference), V3 is the volume of water (15 mL) used during photothermal catalysis, n is the concentration

of diluted DMSO (1.413 mM), V2 is the volume of DMSO (20 μL), V1 is the volume of the reaction solution tested (400 μL), t is reaction time and m is the mass of catalyst.

To make the reader better understand the process, we've modified our description and made it clearer in the revised version.

Comment 14: *The authors emphasized the light response range of Cu-CSCO HNA from visible to NIR. Besides testing with 400 nm cutoff filter, did the authors study the performance of the catalyst under NIR light irradiation?*

Response: Thanks for the Reviewer's concerns. We didn't study the performance under the pure NIR light irradiation before. According to the Reviewer's suggestion, we then tried the NIR light-driven CO_2 reduction using the Cu-CSCO HNA system. As shown in Fig. N21, we found that only small amounts of methane (7.32 $\mu\text{mol g}^{-1} \text{h}^{-1}$) and CO (0.54 $\mu\text{mol g}^{-1} \text{h}^{-1}$) were produced, with almost no liquid product under pure NIR light irradiation. Considering that the low-energy NIR light can't excite the charge carrier by the interband electron transfer in Cu_2O compound, we speculate the trace generation of CH_4 and CO could be induced by the intraband electron transfer in Cu_2Se nanosheet and the heat produced by the NIR light.

Fig. N21 | The performance of Cu-CSCO HNA for CO_2 reduction under NIR light irradiation. (a) The illumination spectrum of NIR light simulator with 800 nm cutoff filter. (b) Liquid products detected by ^1H NMR spectrum and (c) gas products detected by GC after 10 h photocatalysis. (d) The generation rate of various products over this catalytic system

REVIEWERS' COMMENTS

Reviewer #1 (Remarks to the Author):

The authors have met my requirements and this manuscript can be accepted by NC.

Reviewer #2 (Remarks to the Author):

All issues have been well addressed point by point. It can be accepted for publication in Nature Communications.

Reviewer #3 (Remarks to the Author):

My concerns have been addressed and the paper can be accepted now.

Responses to the Reviewers' comments to the manuscript: NCOMMS-24-07495A. We would like to thank all the reviewers for the insightful comments and suggestions, and for their time in helping us to improve this manuscript.

Point-to-point Response to Reviewer #1

Overall comments: *The authors have met my requirements and this manuscript can be accepted by NC.*

Overall response: We greatly appreciate the reviewer for agreeing to recommend our article for publication.

Point-to-point Response to Reviewer #2

Overall comments: *All issues have been well addressed point by point. It can be accepted for publication in Nature Communications.*

Overall response: We thank the Reviewer for agreeing to recommend our article for publication.

Point-to-point Response to Reviewer #3

Overall comments: *My concerns have been addressed and the paper can be accepted now.*

Overall response: We thank the Reviewer for agreeing to recommend our article for publication.